# Fungi Diversity in $PM_{2.5}$ and $PM_1$ at the summit of Mt. Tai: Abundance, Size Distribution, and Seasonal Variation

Caihong Xu[1], Min Wei[1,a], Jianmin Chen[1,2,3,*], Chao Zhu[1], Jiarong Li[1], Ganglin Lv[1],

Xianmang Xu[1], Lulu Zheng[2], Guodong Sui[2], Weijun Li[1], Bing Chen[1], Wenxing

Wang[1], Qingzhu Zhang[1], Aijun Ding[3], Abdelwahid Mellouki[1,4]

[1] Environment Research Institute, School of Environmental Science and Engineering, Shandong University, Ji'nan 250100, China

[2] Shanghai Key Laboratory of Atmospheric Particle Pollution and Prevention (LAP[3]), Fudan Tyndall

Centre, Department of Environmental Science & Engineering, Fudan University, Shanghai 200433, China

[3] Institute for Climate and Global Change Research, School of Atmospheric Sciences, Nanjing University, Nanjing 210023, Jiangsu, China

[4] Institut de Combustion, Aérothermique, Réactivité et Environnement, CNRS, 45071 Orléans cedex 02,

France

[a] now at: College of Geography and Environment, Shandong Normal University, Ji'nan 250100, China

* Corresponding author

E-mail address: jmchen@sdu.edu.cn or jmchen@fudan.edu.cn. (J. M. Chen)

**Abstract.** Fungi are ubiquitous throughout the near-surface atmosphere, where they represent an

20 important component of primary biological aerosol particles. This study combined internal transcribed spacer region sequencing and quantitative real-time polymerase chain reaction (qPCR) to investigate the ambient fungi in fine ($PM_{2.5}$, 50% cutoff aerodynamic diameter $D_{a50}$=2.5 μm, geometric standard deviation of collection efficiency $\sigma_g$=1.2) and submicron ($PM_1$, $D_{a50}$=1 μm, $\sigma_g$=1.2) particles at the summit of Mt. Tai located in the North China Plain, China. Fungal abundance values were $9.4 \times 10^4$

and $1.3 \times 10^5$ copies $m^{-3}$ in $PM_{2.5}$ and $PM_1$, respectively. Most of the fungal sequences were from Ascomycota and Basidiomycota, which are known to actively discharge spores into the atmosphere. The fungal community showed a significant seasonal shift across different size fractions according to Metastats analysis and the Kruskal-Wallis rank sum test. The abundance of *Glomerella* and *Zasmidium* increased in larger particles in autumn, whereas *Penicillium*, *Bullera*, and *Phaeosphaeria* increased in

smaller particles in winter. Three environmental factors, namely, $Ca^{2+}$, humidity, and temperature, were

found to be crucial for the seasonal variation in the fungal community. This study might serve as an important reference for fungal contribution to primary biological aerosol particles.

## 1. INTRODUCTION

Inhaled particulate matter (PM), categorized as $PM_{2.5}$ and $PM_1$ (aerodynamic equivalent diameters of $\leq 2.5$ μm and $\leq 1$ μm, respectively), has been proved to be associated with the increasing morbidity and mortality from cardiovascular and respiratory diseases (Brauer et al., 2013; Wang et al., 2014). Primary biological aerosol particles (PBAPs; about $10^4$–$10^8$ cells cm$^{-2}$) constitute an important component of PM. They can actively metabolize in the atmosphere with their mass concentrations ranging from 5.49 to 102 ng m$^{-3}$ (Zhong et al., 2016). Furthermore, they play an important role in agriculture, the biosphere, cloud formation, global climate, and atmospheric dynamics (Brodie et al., 2007; Despres et al., 2012; Christner et al., 2008; Zhou et al., 2014; Jaenicke et al., 2005). Fungi, the primary group of PBAPs, include 1.5 million unique species, distributed across rural and urban environments (Hawksworth et al., 2001). They actively eject their spores with aqueous jets or droplets into the atmosphere. The global emissions of fungal spores are estimated as the largest source of bioaerosols (Elbert et al., 2007). Pioneering studies have reported global fungal emissions to reach 28 Tg per year and contribute to about 4-13% of the mass concentration of $PM_{2.5}$ (Heald et al., 2009; Womiloju et al., 2003). More recently, some specific fungal species have been verified to be linked with the occurrence of public health problems (Morris et al., 2002; Yadav et al., 2004; Bowers et al., 2012; Bowers et al., 2013; Cao et al., 2014; Ryan et al., 2009). Despite their importance, the abundance, diversity, and community structure of fungi associated with PM have received limited attention in terms of research.

Earlier studies on airborne fungal communities, primarily based on culturing methods, found the dominant phyla to be Ascomycota (AMC) and Basidiomycota (BMC). Some of the species are considered major pathogens and allergens of plants, animals, and humans, e.g., *Hemileia vastatrix*, *Aspergillus*, *Cryptococcus*, and *Pneumocystis* spp. (Despres et al., 2012; Smets et al., 2016). While most of the fungal species remain unknown because cultivable species (typically less than 100) occupy only a tiny minority of all existing species, advances in nucleic acid sequencing allow the accurate determination of both cultured and uncultured microbial communities in environmental samples. For bacterial community composition, Xu et al. (2017a) investigated the abundance and community of bacteria in submicron particles during severe haze episodes in Jinan, China. Later, they discussed the diurnal variation of diverse bacterial communities in cloud water at Mt. Tai, China (Xu et al., 2017b). For diverse fungi in Mainz, Germany, Frohlich-Nowoisky (2009) described the fungal community in coarse (>3 μm) and fine (≤3 μm) PM using internal transcribed spacer (ITS) region sequencing. Yamamoto (2012) reported the crucial influence of aerodynamic diameter and season on the fungal taxonomic composition in the northeastern United States by 454 pyrosequencing. The fungal allergens clustered in the largest size ranges (>9 μm) in the fall season, whereas the pathogens were most abundant in the spring season and were typically observed in particles with aerodynamic diameters of <4.7 μm. Subsequently, DeLeon-Rodriguez (2013) discussed the effect of tropical storm or hurricane

periods on the shift of airborne fungal species over the upper troposphere. Gou et al. (2016) described the fungal abundance and taxonomic composition of fungi in $PM_1$ and $PM_{10}$ in winter in China by 18S rRNA gene sequencing. However, that study focused on the fungal communities in total suspended particles (TSP), $PM_{10}$, and $PM_{2.5}$, and was primarily conducted over the ground's surface; therefore, fungal populations in $PM_1$ at high-elevation sites were not well accounted for. Diverse microbes at high altitudes (such as in cloud water and precipitation) can act as nucleating agents for cloud and ice condensation, influence precipitation patterns (Xu et al., 2017b; Pratt et al., 2009; Creamean et al., 2013; Bower et al., 2013), and drive the biogeochemical cycling of elements in ecosystem processes. Hence, it is essential to advance the knowledge of microbes in PM, especially across the East Asian region which are frequently ravished by dust, haze or other weather phenomenon. During 2013, 2014, and 2015, serious air pollution events associated with the inadequate use of clean energy in the transport, domestic, and industrial sectors affected Northern China, which includes several areas with severe air pollution, namely, Beijing, Tianjin, Shijiazhuang, Jinan, and Qingdao. Most researchers focus attentions on the case study of bacterial abundance and diversity (Gao et al., 2017a; Xu, et al., 2017a; Wei et al., 2017; Cao et al, 2014). Because the various physical, chemical and biological factors caused by the severe haze or dust episodes may shifts on the bacterial community structure. Moreover, the airborne microbial abundance and diversity is also effected by season and meteorological factors, However the investigations on the seasonal variation of fungal characteristics in aerosol particles have been very limited.

Mt. Tai (36°15′N, 117°06′E, 1534 m above sea level), the highest site in the North China Plain, is a tilted fault block mountain, its height increasing from the north to the south, facing the Japanese Islands, Korean Peninsula, East China Sea, and Yellow Sea. The vegetation cover is 80%, with nearly 1000 kinds of plants growing in this area. The number of tourists, from both China and abroad, visiting this mountain increased from 5.5 million in 2014 to 5.9 million in 2015. Past investigations in this region mainly concentrated on the physicochemical characteristics of aerosol particles and cloud water and their influence on air quality and human health. Thus far, there have been no studies addressing the diverse fungal community in aerosol particles at Mt. Tai, necessitating the development of a reliable knowledge base on the atmospheric aerosols in such scenic destinations.

The objectives of the present study were: (i) to fill the knowledge gaps regarding the ambient fungi of $PM_{2.5}$ and $PM_1$ at a high-elevation site of East Asia, (ii) to elucidate the size-based differences between the data of ambient fungal concentration and viable fungal community structure at different taxonomic levels across different seasons, and (iii) to estimate whether environmental factors play a role in the variation of fungal characteristics at Mt. Tai.

## 2. MATERIALS AND METHODS

### 2.1 Sample collection

At Mt Tai, spring occurs from March to May; summer, June to August; fall, September to November; and winter, December to February, according to the environmental temperature. Two middle-volume (100 L min$^{-1}$) samplers (TH-150A; Wuhan Tianhong Instruments Co. Ltd., Wuhan, China) were deployed with particles larger than 2.5 μm and 1μm trapped by the impactors and particles smaller than 2.5 μm and 1 μm collected on the quartz filters, respectively. The 50% cutoff aerodynamic diameter are 2.5μm and 1μm, respectively. The smaller the aerosol particles, the higher collection efficiency. Sixty quartz membrane filters (PALL, NY, USA., 88 mm) were obtained for 23 h (9:00 am to 8:00 am the next day) over 8–13 days during each season from 2014 to 2015 at the summit of Mt. Tai (Table 1). The blank filters were obtained by placing sterilized quartz microfiber filters inside the sampler without any operation. Before sampling, all the filters were baked in a muffle furnace at 500 ℃ for 5 h, placed into sterilized aluminum foil, and then deposited into a sealed bag. To avoid contamination, the sampling filter holder and materials used for changing filters were treated with 75% ethanol every day. After sampling, the samples were stored at −80°C until the next analysis. PM$_{2.5}$ and PM$_1$ mass concentrations were monitored by a synchronized hybrid ambient real-time particulate monitor (Model 5030; Thermo Fisher Scientific, Wilmington, DE, USA). Half of the PM$_{2.5}$ and PM$_1$ filters were used to analyze water-soluble inorganic ions (NO$_3^-$, SO$_4^{2-}$, NH$_4^+$, K$^+$, Ca$^{2+}$, Na$^+$, and Mg$^{2+}$) by an ambient ion monitor (URG-9000; UGR Corporation, Chapel Hill, NC, USA). The remaining filters were analyzed in the same batch of laboratory experiments, including DNA extraction, PCR amplification, quantitative real-time PCR (qPCR), and Illumina sequencing, except for sample A29 in December 9, 2014 (accidentally omitted in the first batch of Illumina sequencing). Considering that a part of the sequences in the 2 batches of experiments differed, we removed this sample before quality control. Meteorological data, including relative humidity, wind speed, wind direction, and temperature, were obtained from http://www.underground.com at a resolution of 3 h during the sampling period. The visibility was monitored online by a visibility sensor (Model PWD22; Vaisala, Finland) with a maximum limit of 20 km.

### 2.2 DNA extraction and PCR amplification

The sample pretreatment and DNA extraction experiments were performed following an protocol optimized by Jiang et al., (2015). This protocol can extract sufficient DNA from low-biomass environmental samples (e.g., aerosol particles, and other alike) and boosted the DNA extraction efficiency more than twice better than the non-optimized extraction method. Besides, it has been applied for studying airborne microbial diversity in different environment (Cao et al., 2014; Deng et al., 2016; Tong et al., 2017; and Gao et al., 2017b). Half of the filters (about 121.64 cm$^2$) were cut into small pieces, inserted into 50-mL Falcon tubes filled with sterilized 1 × PBS buffer, and centrifuged at 200 ×$g$ for 3 h at 4 ℃. The resuspension was collected into a 0.2-μm Supor 200 PES membrane disc filter. We cut the PES membrane disc filter into small pieces, heated the pieces to 65 ℃ in PowerBead

tubes for 15 min, and then vortexed them for 15 min. DNA was extracted according to the standard PowerSoil DNA isolation protocol (Judd et al., 2016) and purified by AMPure XP bead purification. A parallel extraction procedure was performed with the blank filter to check for sample contamination. DNA concentrations were quantified by a NanoDrop 2000 spectrophotometer (Thermo Fisher Scientific). The fragments of ITS1 regions were amplified from genomic DNA by PCR using the forward primer ITS1F (5′-CTTGGTCATTTAGAGGAAGTAA-3′) and the reverse primer ITS4 (5′-TCCTCCGCTTATTGATATGC-3′), which target the fungal ITS region of the rRNA gene (Manter et al., 2007). The experiment was conducted using the Gene Amp® PCR System 9700 (Applied Biosystems, CA, USA) in a total volume of 50 μL PCR mix containing PCR buffer (1×), 1.5 μM $MgSO_4$, 0.4 μM of each deoxynucleotide triphosphate, 0.3 μM each of the forward and reverse primers, 0.5 U Ex Taq (TaKaRa, Dalian, China), 100 ng template DNA, and double distilled $H_2O$. The thermal cycling profile was 94 ℃ for 1 min; 35 cycles of denaturation at 98 ℃ for 20 s, annealing 68 ℃ for 30 s, and elongation at 72 ℃ for 45 s; and final extension at 72 ℃ for 5 min. Three replicates of PCR for each sample were combined together. The final products were separated by 1.5% agarose gel electrophoresis and purified using the Qiaquick PCR purification kit (Qiagen, Valencia, CA, USA ). Purified amplicons were quantified by a Qubit 2.0 fluorometer (Thermo Scientific) and pooled with equal molar amounts. Sequencing libraries were generated using the Truseq DNA PCR-Free Sample Prep Kit following manufacturer's instructions. Sequencing was performed on an Illumina MiSeq instrument (Illumina, San Diego, CA, USA) with the MiSeq reagent kit V3 (Illumina) according to the standard protocols.

## 2.3 Sequence analyses

After high-throughput sequencing, we removed the chimeric and low-quality sequences using the FASTX-ToolKit (http://hannonlab.cshl.edu/fastx_toolkit) and UCHIME algorithm (Edge et al., 2011) before diversity analysis and statistical analysis. The remaining high-quality sequences were normalized to 7973 reads to compare the different samples effectively. They were then clustered into operational taxonomic units (OTUs) at a 97% similarity cutoff using USEARCH software (version 7.1, http://drive5.com/uparse/). We used the OTUs as the basis for estimating the alpha diversity and beta diversity. The taxonomy of ITS sequences was analyzed by RDP Classifier against the Unite database (Release 7.0, http://unite.ut.ee/index.php; Koljalg et al., 2013) using a confidence threshold of 70%. RDP Classifier was used to determine the taxonomic composition at the phylum, class, order, family, genus, and species levels (Koiv et al., 2015; Miettinen et al., 2015). Alpha diversity estimators, including Chao1, Simpson's index, and Shannon's index, were calculated by the Quantitative Insights into Microbial Ecology software (version 1.8.0, http://qiime.org/scripts/assign_taxonomy.html; Kuczynski et al., 2011). The raw reads were deposited into the NCBI Sequence Read Archive database under accession number SRR5146156.

## 2.4 qPCR for ITS regions

To determine the fungal biomass, we performed qPCR (Gao et al., 2017a; Yamaguchi et al., 2016; Lee et al. 2010) using a CFX96 real-time PCR detection system (Bio-Rad, Hercules, CA, USA) in 25-μL reaction mixtures containing 12.5 μL TransStart Green qPCR SuperMix, 1 μL ITS3-KYO2 (5′-GATGAAGAACGYAGYRAA-3′), 1 μL ITS4 (5′-TCCTCCGCTTATTGATATGC-3′), 5 μL sample DNA, and 5.5 μL double-distilled H$_2$O. The amplification followed a three-step PCR for fungal ITS regions: 40 cycles of denaturation at 95 ℃ for 30 s, primer annealing at 52 ℃ for 30 s, and extension at 72 ℃ for 30 s. A standard curve was created using tenfold dilution series of fungal ITS region plasmids. Assuming that the average fungal genome has about 30–200 rRNA copies, the fungal concentrations were calculated using the methods described by Lee et al. (2010) and van Doorn et al. (2007).

### 2.5 Fungal contribution to atmospheric organic carbon

The contributions of fungal spores to organic carbon (OC) were calculated using mannitol as a biotracer. We assumed 1.7 pg mannitol and 13 pg OC per spore. To assess the contribution of fungal spores to the OC and to the mass balance of atmospheric aerosol particles quantitatively, we used the weighted-average carbon conversion factor of 13 pg C per spore and of 33 pg fresh weight per spore, which had been obtained earlier as the average carbon content of spores from airborne fungal species (Bauer et al., 2008; Zhu et al., 2016; Liang et al., 2017).

### 2.6 Statistical analyses

To determine the differences in the fungal community variations among different size fractions, meta-analyses based on the permutation $t$-test were conducted using Mothur software (version 1.35.1). The program Metastats can produce a tab-delimited table to display the mean relative abundance of the mean, variance, and standard error, together with the $p$-values and $q$-values. Values were considered significant if $p \leq 0.05$ and $q \leq 0.05$. The Kruskal-Wallis rank sum test was used to evaluate the seasonal variation of the microbial community. Boxplots and $q$-values are have been provided for illustration. The relationship between the ambient microbial concentrations and environmental factors, including PM concentrations and chemical compositions, was assessed with nonparametric Spearman's rank correlation coefficients by SPSS 16.0.

### 3. RESULTS AND DISCUSSION

### 3.1 Concentration of fungal spores in PM$_{2.5}$ and PM$_1$

PM$_{2.5}$ and PM$_1$ samples were collected during summer, autumn, and winter at the summit of Mt. Tai. Temporal variations of the mass concentration and corresponding fungal spore numbers of PM$_{2.5}$ and PM$_1$ are summarized in Table 1. PM$_1$ mass concentration was stable over different seasons, while PM$_{2.5}$ demonstrated a high seasonal variation, with higher concentrations in summer (44.7 μg m$^{-3}$) than in autumn (37.2 μg m$^{-3}$) and winter (21.7 μg m$^{-3}$). The values were much lower than that in the summer of 2006 (123.1 μg m$^{-3}$; Deng et al., 2011) and comparable with that in the summer of 2007

(59.3 μg m$^{-3}$; Zhou et al., 2009). The average PM$_1$/PM$_{2.5}$ ratios were 0.45 in summer, 0.65 in autumn, and 0.84 in winter, implying that fine particles dominated in summer, while submicron particles dominated in autumn and winter.

qPCR revealed an average fungal gene copy number of 9.4 $\times 10^4$ copies m$^{-3}$ (ranging from 1.0 $\times 10^4$ to 4.8 $\times 10^5$ copies m$^{-3}$) and 1.3 $\times 10^5$ copies m$^{-3}$ (ranging from 3.7 $\times 10^3$ to 1.0 $\times 10^6$ copies m$^{-3}$) in PM$_{2.5}$ and PM$_1$, respectively. There is no significant differences between PM$_{2.5}$ and PM$_1$ based on the uncertainty estimate (95% confidence intervals). Assuming an average rRNA gene copy number of 200 per fungal genome (van Doorn et al., 2007; Lee et al., 2010), we obtained an average fungal concentration of 467 spores m$^{-3}$ and 644 spores m$^{-3}$ in PM$_{2.5}$ and PM$_1$, respectively. The concentrations at Mt. Tai were lower than those at surface ground sites, including those in Korea (ranging from 9.56 $\times$ 10$^1$ to 4.2 $\times 10^4$ cells m$^{-3}$; Lee et al., 2010), Austria (1.8 $\times 10^4$ cells m$^{-3}$ in urban sites and 2.3 $\times 10^4$ cells m$^{-3}$ in suburban sites; Bauer et al., 2008), Portugal (ranging from 891 to 964 spores m$^{-3}$; Oliveira et al., 2009), and the United States (6450 spores m$^{-3}$; Tsai et al., 2007). Our lower values might be ascribed to an underestimation of the fungal numbers. We used a higher gene copy number of 200 for each microbe studied, whereas DeLeon-Rodriguez et al. (2013) employed a lower number of rRNA copies of fungal genomes (30-100 copies per genome). The discrepancy between our results and those of Lee et al. (2010) might be because of the differences in sample type, sampling time, and altitude. Lee et al. (2010) focused on the fungal concentration in TSP by a high-volume TSP sampler (0.225 m$^3$ min$^{-1}$) 15 m above the ground in autumn and winter, whereas we obtained the PM$_{2.5}$ and PM$_1$ by middle-volume samplers (0.1 m$^3$ min$^{-1}$) 1534 m above the ground in summer, autumn, and winter. It is difficult to explain the disparity between different studies without uniform guidelines for the sampling and quantitative assessment of bioaerosols.

Fungal abundance varied seasonally with different size particles in the near-surface atmosphere. Saari et al. (2015) found that coarse fluorescent bioaerosol particles (1.5-5 μm) increased in summer, whereas in winter, these particles primarily existed in smaller particles (0.5-1.5 μm). The snow cover and decreased biological activity in winter resulted in the disappearance of microbes from the coarse fluorescent bioaerosol particles. In this study, the highest fungal concentration in PM$_{2.5}$ was observed in summer (641 spores m$^{-3}$), whereas the highest value in PM$_1$ was found in autumn (1033 spores m$^{-3}$), indicating different origins of fungal spores. Huffman et al. (2010) found that long-range transport of aerosols and anthropogenic sources such as combustion influence the fluorescent biological aerosol particles having diameters less than 1 μm. During the autumn sampling, no obvious straw combustion phenomena occurred, and we detected some long-range transportation events in November 2014. Long-range transported airborne PM were mainly derived from the outer Mongolia regions, well-known to be one of the dustiest places in East Asia (November 6), Siberia (November 3 and November 12), and Taklimakan and Gobi desert regions (November 5). Influenced by the air movements from the desert region, the corresponding fungal abundance increased from 6.18 $\times 10^4$ to 103 $\times 10^4$ copies m$^{-3}$ (about 16.7-folds). Similarly, the corresponding fungal abundance influenced by air parcels from Siberian regions increased to 22.2 $\times$ 10$^4$ and 18.3 $\times$ 10$^4$ copies m$^{-3}$, respectively. Hence, we

hypothesized that the long-range transport of air parcels from north China might have contributed to the fungal enrichment of $PM_1$. In addition, the increased fungal abundance might be explained by meteorological diversity (Abdel Hameed et al., 2012). Low wind speed hinders fungal dispersal owing to the accumulation effect. According to Almaguer et al. (2014), in Cuba, the calm winds coming from the southwest direction induce the accumulation of fungal spores over the northern coast of the island. Lin et al. (2000) observed a strongly negative correlation between wind speeds of $< 4$ m s$^{-1}$ and fungal concentration; the fungal concentration increased as the wind speed became higher than 5 m s$^{-1}$ in the Taipei area. In our present study, the fungal abundance in $PM_1$ showed no obvious increase under breezy conditions (wind speed $< 2$ m s$^{-1}$) mainly from the southern direction (Figure 1). When the wind speed was higher than 2 m s$^{-1}$, the fungal abundance increased markedly under the influence of westerly winds. As the westerly wind velocity increased, the fungal concentration increase slowly. Meanwhile, in $PM_{2.5}$, the fungal abundance increased with wind velocities higher than 2 m s$^{-1}$, mainly from the northwest direction of the continental areas, where diverse vegetation grows. The phenomenon implies that westerly and northwesterly winds might highly induce fungal growth and abundance in PM at Mt. Tai.

**3.2 Contribution of spores to OC concentrations and PM mass**

OC, accounting for 7-80% of PM mass, constitutes a significant fraction of atmospheric aerosols (Yu et al., 2004; Ram et al., 2012; Ho et al., 2012). Ambient fungi are considered a possible source of OC in PMs. Cheng et al. (2009) estimated the mean fungal OC concentrations in Hong Kong to be 3.7, 6.0, and 9.7 ng m$^{-3}$, corresponding to 0.1%, 1.2%, and 0.2% of the total OC in $PM_{2.5}$, $PM_{2.5-10}$, and $PM_{10}$, respectively. In the present study, the range and average concentrations of fungal contribution to atmospheric OC and mass concentration $PM_{2.5}$ and $PM_1$ are listed in Table 1. The daily averaged concentrations of fungal OC in $PM_{2.5}$ and $PM_1$ were 6.1 and 8.3 ng C m$^{-3}$, respectively, with the respective contributions to PM being 0.067% and 0.096%, indicating that airborne fungal spores as a minor source of carbonaceous aerosols cannot be ignored at Mt. Tai. The fungal contribution to OC obtained at Mt. Tai was comparable with that observed at an urban site in Hong Kong (3.7 ng C m$^{-3}$; Cheng et al., 2009) but lower than that obtained at an urban site in Austria (117.9 ng C m$^{-3}$; Bauer et al., 2008) and a forest site on Hainan Island (147-923 ng C m$^{-3}$; Zhang et al., 2015). The discrepancy between the abovementioned studies can be justified by the difference in particle type studied (TSP, $PM_{10}$, $PM_{2.5}$, and $PM_1$), fungal concentration, spore carbon content, and assessment method (e.g., sugar alcohol, cultivation, mannitol, and light microscopy). On the basis of the same conversion factor of 13 pg C spore$^{-1}$ by mannitol, the results were much lower than that obtained at an urban site in Beijing (0.3 $\pm$ 0.2 $\mu$g C m$^{-3}$; Liang et al., 2017), implying a lower fungal concentration at Mt. Tai than that in Beijing. More studies are needed to better understand the spatial, temporal, and size distributions of fungal OC contributions to atmospheric particles in urban areas in the North China Plain.

### 3.3 Taxonomic diversity and composition of ambient fungi

On average, 509 and 475 OTUs were obtained in $PM_{2.5}$ and $PM_1$, respectively, which were higher than those obtained in earlier airborne fungal studies at the ground level in Beijing, China (34-285; Yan et al., 2016) and Rehovot, Israel (121-178; Dannemiller et al., 2014). The OTUs associated with $PM_{2.5}$ in summer, autumn, and winter were higher than those associated with $PM_1$, implying more diverse fungal spores in $PM_{2.5}$. However, the Shannon and Chao1 indices showed different trends in $PM_{2.5}$ and $PM_1$ (Figure 2). The ambient fungi showed the highest richness and diversity in winter, followed by autumn and summer. Although $PM_1$ mass concentration dominated in autumn and winter, the corresponding fungal diversity was lower than that in $PM_{2.5}$. Similarly, the dominant $PM_{2.5}$ mass concentration in summer presented lower diversity than that in $PM_1$.

In the fungal community, AMC (89.7%) and BMC (7.0%) were the predominant phyla, and they are known to actively discharge spores into the atmosphere (Figure 3A). The remaining phyla were Zygomycota (ZMC) and Glomeromycota. AMC and BMC present a global pattern across continental (Austria, Arizona, Brazil, and Germany), coastal (Taiwan, Puerto Rico, and UK), and marine sites (Pacific, Indian, Atlantic, and Southern Ocean) (Frohlich-Nowoisky et al., 2012). In continental samples, BMC (64%) seems to be more abundant than AMC (34%), whereas in marine sites, AMC (72%) is about 2.6 times more abundant than BMC. Herein, the abundance of AMC was approximately 12.8 times higher than that of BMC. Members of AMC have single-celled or filamentous vegetative growth forms that are easily aerosolized, unlike BMC (Womack et al., 2015). Furthermore, 10 classes belonging to AMC, 10 to BMC, and 1 to ZMC were observed (Figure 3B). The preponderant classes belonging to AMC were Dothideomycetes (37.3%), Sordariomycetes (15.0%), and Eurotiomycetes (6.1%). The dominant orders in Dothideomycetes included Pleosporales (14.9%), Capnodiales (5.3%), and Botryosphaeriales (1.6%) (Figure 3C). Pleosporales has been reported to include fungi allergenic to local residents (Rittenour et al., 2014). The values were lower than those reported in Beijing's PM (Pleosporales: 29.39% and Capnodiales: 27.96%) (Yan et al., 2016). Likewise, the dominant classes in BMC were Agaricomycetes (4.4%) and Tremellomycetes (1.5%), including the orders Polyporales (2.5%), Agaricales (1.6%), and Tremellales (1.2%). About 291 taxa from the genus level were determined, including *Alternaria*, *Glomerella*, *Zasmidium*, *Pestalotiopsis*, *Aspergillus*, and *Phyllosticta*. The distribution was discrepant with that at the ground level, wherein *Cladosporium* occupied more than 50% of total fungi, followed by *Alternaria*, *Didymella*, and *Khuskia* (Oh et al., 2014). The top 5 orders (Pleosporales, Xylariales, Eurotiales, Capnodiales, Polyporales) and genera (*Alternaria* and *Aspergillus*) were commonly observed in suspended aerosol particles (including TSP, $PM_{10}$, $PM_{2.5}$, and $PM_1$) but showed variable relative abundances, as shown in Table 2. We attribute this disparity to the different sampling approaches, instruments, and analysis methods. This aspect needs to be probed and studied in depth in the future.

### 3.4 Implication of the allergenic and pathogenic fungi

To date, about 123 fungal genera (mainly belonging to the phylum AMC) have been identified to be human allergens (Simon-Nobbe et al., 2008). Of the 11 potentially allergy-inducing AMC species and 1 potentially allergy-inducing BMC species found at Mt. Tai, the 3 most common species were *Aspergillus flavus*, *Blumeria graminis*, and *Saccharomyces cerevisiae*. *Aspergillus flavus* is a common human pathogen found in air, and it is also a human allergen and mycotoxin producer (Adhikari et al., 2004). It is associated with invasive aspergillosis and superficial infections (Hedayati et al., 2007). *Blumeria graminis*, found on the surface of plant leaves, causes powdery mildew on cereal plants (Belanger et al., 2003). Such pathogens and allergens are expected to be widely spread around the atmospheric environment in temperate and tropical zones (Vermani et al., 2010). Our results also revealed that the abundance of potential allergenic and pathogenic fungal spores in summer were the highest compared to those in autumn and winter. Clinicians should consider the fungal spores described herein as a possible cause of human and plant disease attacks under long exposure to airborne particles throughout the year, especially in the summer season. Furthermore, the abundance of the abovementioned allergenic and pathogenic fungal spores in $PM_1$ was about 3.8 times higher than that in $PM_{2.5}$ in summer, implying relatively higher health risks of smaller particles. Residents and even visitors at Mt. Tai should be warned about this phenomenon.

### 3.5 Size distribution and seasonal variation of fungal communities

Both fungal abundance and fungal community show a seasonal trend across different size fractions (Awad et al., 2013). Yamamoto et al. (2012) observed that the pathogenic fungi were mainly detected at $PM_{4.7}$ (PM with aerodynamic diameter < 4.7 μm), while the allergenic fungi existed primarily at PM with aerodynamic diameter > 9 μm. In the present study, a discrepant size distribution of the fungal community was observed according to the Metastat analysis by permutation *t*-tests (Table 3). *Glomerella*, *Zasmidium*, and *Phyllosticta* were abundantly enriched in $PM_{2.5}$, while the abundance of *Preussia*, *Truncatella*, *Umbelopsis*, *Sebacina*, and *Cordyceps* increased in $PM_1$. The Kruskal-Wallis rank sum test (Figure 4) showed that 6 fungal genera had apparent seasonal variation. *Glomerella* and *Zasmidium* increased in autumn and decreased as the particle size increased. *Glomerella* was widely found on the surface of leaves, suggesting that leaf senescence is an important source of fungi in $PM_{2.5}$ in autumn (Wang et al., 2015). Some crucial environmental factors having a potential influence on fungal release and growth, such as temperature; $NO_2$; $PM_{10}$; $SO_2$; CO; relative humidity (Yan et al., 2016); radiation, vegetation, and urbanization, and accidental events, e.g., dust storms (Prospero et al., 2005), rainfall (Zhang et al., 2015), hurricanes (DeLeon-Rodriguez et al., 2013), and haze (Yan et al., 2016), (Moreau et al., 2016), have been identified. In the current study, Spearman's rank coefficient analysis indicated that $Ca^{2+}$, a typical water-soluble inorganic ion from dust, was negatively related to the prevalence of *Glomerella* and *Zasmidium* in autumn (Figure 5). In winter, the abundance of *Penicillium*, *Bullera*, and *Geosmithia* increased owing to their sensitivity to low temperature (Sousa et al., 2008; Abdel Hameed et al., 2012). The results based on Spearman's rank correlation test analysis

support this notion (Figure 5, $p < 0.01$). Humidity, another important factor for fungal release into the atmosphere either by active or passive modes, is a crucial factor for the variation in fungal spores such as *Lophium* ($p < 0.01$), *Cenococcum* ($p < 0.05$), *Tricholoma* ($p < 0.05$), and *Candida* ($p < 0.05$). In summer, no distinct difference was observed based on the top 40 fungal genera (Figure 4). However, some trace fungal genera presented inverse correlation with temperature (*Coccomyces*, $p < 0.01$; and *Dictyosporium*, $p < 0.01$) and humidity (*Botryosphaeria*, $p < 0.001$; *Coccomyces*, $p < 0.01$; and *Dictyosporium*, $p < 0.01$). We determined that 4 crucial environmental factors contributed to the variation in the fungal community. As culture studies on the detailed mechanism of the effects of environmental factors on specific fungal spores have been limited, this problem still needs to be surveyed over a longer duration. Moreover, the detailed relationship of bioaerosols with environmental factors require further study.

**4. CONCLUSIONS**

Diverse airborne fungal spores is relevant for studies on the atmosphere, biogeoscience, climate and ecology, environmental hygiene, agriculture, and bioengineering. As the details of fungal spores present at high-elevation sites remain unknown, the detection and characterization of ambient fungi can help elucidate the regional and global distribution of diverse fungi. Herein, we provide a comprehensive framework of the fungal abundance and communities associated with different seasons across $PM_{2.5}$ and $PM_1$ at Mt. Tai. The results revealed that the concentration and fungal community structure at Mt. Tai difference considerably from those reported for surface ground sites. Over the sampling period, average fungal concentrations of 467 spores $m^{-3}$ and 644 spores $m^{-3}$ in $PM_{2.5}$ and $PM_1$ were calculated. In addition to long-distance air mass movement events, westerly and northwesterly winds also favor the increase in fungal abundance. Diverse fungal communities presented significant seasonal variation across different size particles. The prevalence of *Glomerella* and *Zasmidium* increased in autumn and decreased as the particle size increased. In winter, the prevalence of *Penicillium*, *Bullera*, and *Geosmithia* increased with the decrease in particle size. No distinct disparity was observed in summer. The variation in fungal profile can be influenced by environmental factors, including humidity, temperature, wind speed, $PM_{2.5}$, and some chemical components in PMs (including $Ca^{2+}$). Nevertheless, the detailed specific effects of environmental factors on the ambient fungal community remain poorly explained. In further studies, a combination of traditional culture-based methods and metagenomics may help answer the various unresolved questions.

**ACKNOWLEDGEMENTS**

This work was supported by the National Natural Science Foundation of China (No. 41375126, 21527814), Taishan Scholar Grant (grant number ts20120552), Cyrus Tang Foundation (No. CTF-FD2014001), the Ministry of Science and Technology of of China (No. 2016YFC0202701,

2014BAC22B01) , and the European Union's Horizon 2020 research and innovation programme under grant agreement No. 690958 (MARSU-Project).

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

**Table and Figure Captions**

Table 1. Sample descriptions and the associated meteorological characteristics of the atmosphere, including the temperature, relative humidity, visibility, $PM_{2.5}$ mass concentration, $PM_1$ mass concentration, and fungal cell concentrations on the basis of qPCR analysis of SSU rRNA gene copy numbers in $PM_{2.5}$ and $PM_1$.

Table 2. The relative abundance of the top 5 orders and 2 genera in TSP, $PM_{10}$, $PM_{2.5}$, and $PM_1$ (RAS[a] indicates relative abundance in submicron particles, and RAF[b] indicates relative abundance in fine particles).

Table 3. Metastats analysis showing the fungal genera that are significantly different among $PM_{2.5}$ and $PM_1$.

Figure 1. Relationships between fungal number concentrations of $PM_{2.5}$ and $PM_1$ with wind speed and wind direction.

Figure 2. Statistical comparisons of OTUs, and Chao1 and Shannon indices among summer, autumn, and winter in $PM_{2.5}$ and $PM_1$.

Figure 3. Relative abundances of fungal communities of $PM_{2.5}$ and $PM_1$ at phylum (A), class (B), and order level (C).

Figure 4. Variance analysis of fungal genera based on the Kruskal-Wallis rank sum test.

Figure 5. Heatmap analysis of the top 64 fungal genera based on Spearman's rank correlations (\*\*\*$p < 0.001$; \*\*$p < 0.01$; \*$p < 0.05$). Red arrows indicate that the specific fungi varied significantly in different seasons.

**Table 1.**

| Season | Date of Collection | T | RH | PM$_{2.5}$ | | | | PM$_1$ | | | |
|---|---|---|---|---|---|---|---|---|---|---|---|
| | | | | MC | Fungal SSU rRNA gene copy number | Fungal spore OC MC | Fungal spore MC | MC | Fungal SSU rRNA gene copy number | Fungal spore OC MC | Fungal spore MC |
| Unit | | °C | % | μg/m³ | E+04 copy/m³ | ng C /m³ | μg /m³ | μg/m³ | E+04 copy/m³ | ng C /m³ | μg /m³ |
| Summer | 06/25/15 | 12.6 | 98 | 5.5 | 11.00 | 7.15 | 0.02 | BDL | 18.00 | 11.69 | 0.03 |
| | 06/26/15 | 14 | 97 | 2.8 | 3.58 | 2.33 | 0.01 | BDL | 6.51 | 4.23 | 0.01 |
| | 06/27/15 | 15.1 | 94.6 | 52.7 | 1.01 | 0.66 | 0.00 | 18.1 | 0.40 | 0.26 | 0.00 |
| | 06/28/15 | 16.9 | 84.4 | 91.1 | 6.85 | 4.45 | 0.01 | 40.0 | 3.10 | 2.02 | 0.01 |
| | 06/29/15 | 17.3 | 62.6 | 16.8 | 4.71 | 3.06 | 0.01 | 13.3 | 2.79 | 1.81 | 0.00 |
| | 07/03/15 | 17.7 | 31.0 | 15.3 | 12.20 | 7.92 | 0.02 | 12.7 | 11.40 | 7.42 | 0.02 |
| | 07/07/15 | 16.9 | 84.4 | 94.0 | 47.70 | 31.00 | 0.08 | 39.9 | 4.20 | 2.73 | 0.01 |
| | 07/08/15 | 17.3 | 62.6 | 110.9 | 22.60 | 14.71 | 0.04 | 42.0 | 5.31 | 3.45 | 0.01 |
| | 08/07/15 | 17.4 | 97.6 | 13.5 | 5.64 | 3.67 | 0.01 | 11.4 | 6.94 | 4.51 | 0.01 |
| Autumn | 10/22/14 | 6.7 | 60.7 | 40.1 | 9.34 | 6.07 | 0.02 | 28.1 | 0.37 | 0.24 | 0.00 |
| | 10/25/14 | 10.3 | 80.7 | 48.1 | 7.28 | 4.73 | 0.01 | 34.6 | 45.70 | 29.73 | 0.08 |
| | 10/26/14 | 11.4 | 73.6 | 50.8 | 6.78 | 4.41 | 0.01 | 31.9 | 8.26 | 5.37 | 0.01 |
| | 11/03/14 | 0.3 | 21.6 | 4.9 | 1.57 | 1.02 | 0.00 | BDL | 22.20 | 14.44 | 0.04 |

| Date | | | | | | | | | |
|------|------|------|------|-------|-------|------|------|--------|--------|-------|
| 11/04/14 | 2.6 | 33.7 | 31.6 | 7.95 | 5.17 | 0.01 | 24.5 | 6.18 | 4.01 | 0.01 |
| 11/05/14 | 4.1 | 30.9 | 33.1 | 3.70 | 2.41 | 0.01 | 25.6 | 103.00 | 66.75 | 0.17 |
| 11/06/14 | 5.1 | 19.3 | 22.7 | 12.70 | 8.24 | 0.02 | 18.0 | 8.86 | 5.76 | 0.01 |
| 11/07/14 | 2.6 | 34.0 | 19.8 | 7.89 | 5.13 | 0.01 | 16.0 | 6.39 | 4.15 | 0.01 |
| 11/08/14 | 2.4 | 45.7 | 22.4 | 14.70 | 9.54 | 0.02 | 17.8 | 2.92 | 1.90 | 0.00 |
| 11/09/14 | 1.1 | 73.1 | 77.1 | 5.56 | 3.61 | 0.01 | 33.5 | 3.61 | 2.34 | 0.01 |
| 11/10/14 | 3.0 | 49.0 | 49.2 | 9.38 | 6.10 | 0.02 | 37.2 | 16.70 | 10.87 | 0.03 |
| 11/11/14 | 2.7 | 65.4 | 32.7 | 27.50 | 17.85 | 0.05 | 25.3 | 26.30 | 17.07 | 0.04 |
| 11/12/14 | 1.0 | 50.1 | 51.7 | 7.50 | 4.87 | 0.01 | 25.7 | 18.30 | 11.87 | 0.03 |
| Winter 12/03/14 | -8.9 | 24.4 | 13.7 | 5.03 | 3.27 | 0.01 | 9.7 | 6.84 | 4.45 | 0.01 |
| 12/04/14 | -11 | 39.1 | 35.0 | 8.68 | 5.64 | 0.01 | 30.6 | 2.78 | 1.81 | 0.00 |
| 12/05/14 | -10.6 | 23.4 | 14.5 | 1.09 | 0.71 | 0.00 | 13.3 | 16.20 | 10.52 | 0.03 |
| 12/06/14 | -5.7 | 11.0 | 9.1 | 6.32 | 4.11 | 0.01 | 8.3 | 4.15 | 2.70 | 0.01 |
| 12/07/14 | -5.4 | 45.7 | 38.8 | 7.90 | 5.14 | 0.01 | 30.9 | 9.36 | 6.08 | 0.02 |
| 12/08/14 | -7.9 | 35.7 | 36.5 | 1.33 | 0.86 | 0.00 | 29.0 | 7.17 | 4.66 | 0.01 |
| 12/09/14 | -5.3 | 16.1 | 16.5 | 10.10 | 6.55 | 0.02 | 13.5 | 5.73 | 3.72 | 0.01 |
| 12/10/14 | -5.6 | 58.3 | 9.3 | 3.24 | 2.10 | 0.01 | 8.1 | 7.10 | 4.62 | 0.01 |

*MC-Mass Concentration *T-Temperature *RH-Relative Humidity *BDL-Below the Detection Line

**Table 2.**

| Taxonomic level | Common Fungi | RAS[a] | RAF[b] | References | Samplers | Sample Type | Concentration or Abundance |
|---|---|---|---|---|---|---|---|
| Genera | Alternaria | 11.7 | 6.2 | Adhikari et al., 2004 | Andersen sampler (Thermo Andersen, Smyrna, 300082-5211, USA) | TSP | 2.6% |
| | | | | Dannemiller et al., 2014 | High volume PM10 samplers (Ecotech, Knoxfield, VIC, Australia) | PM$_{10}$ | >1% |
| | | | | Alghamdi et al., 2014 | PM2.5 samplers (Staplex Air Sampler Division, USA) | PM$_{2.5}$ | 2.6% |
| | | | | Gou et al., 2016 | Low volume air sampler (BGI, USA) | PM$_1$ | >1% |
| | Aspergillus | 2.3 | 1.9 | Cao et al., 2014 | Air samplers (Thermo Electron Corp., MA, U.S.) | PM$_{10}$ and PM$_{2.5}$ | Abundant |
| | | | | Gou et al., 2016 | Low volume air sampler (BGI, USA) | PM$_{10}$ and PM$_1$ | Abundant |
| Order | Pleosporales | 18.4 | 45.4 | Rittenour et al., 2014 | Buck Bioaire Sampler (A.P. Buck, Inc, Orlando, FL, USA) | TSP | 46% |
| | | | | Yan et al., 2016 | Air samplers (Air Metrics, USA, 5 L min$^{-1}$) | PM$_{10}$ and PM$_{2.5}$ | 29.4% |
| | Xylariales | 5.0 | 14.4 | Gou et al., 2016 | Low volume air sampler (BGI,USA) | PM$_{10}$ and PM$_1$ | 10-15% |
| | | | | Womack et al., 2015 | SKC Biosamplers (BioSampler SKC Inc.) | TSP | Abundant |
| | Eurotiales | 4.8 | 13.3 | Gou et al., 2016 | Low volume air sampler (BGI, USA) | PM$_{10}$ and PM$_1$ | 0-5% |
| | | | | Yan et al., 2016 | Air samplers (Air Metrics, USA, 5 L min$^{-1}$) | PM$_{10}$ and PM$_{2.5}$ | 10.6% |
| | Capnodiales | 4.4 | 12.5 | Yan et al., 2016 | Air samplers (Air Metrics, USA, 5 L min$^{-1}$) | PM$_{10}$ and PM$_{2.5}$ | 27.96% |
| | | | | Gou et al., 2016 | Low volume air sampler (BGI, USA) | PM$_{10}$ and PM$_1$ | 10-15% |
| | | | | Gou et al., 2016 | Low volume air sampler (BGI, USA) | PM$_{10}$ and PM$_1$ | ~25% |

| Polyporales | 2.5 | 6.4 | Womack et al., 2015 | SKC Biosamplers (BioSampler SKC Inc.) | TSP | Abundant |
|---|---|---|---|---|---|---|
| | | | Yan et al., 2016 | Air samplers (Air Metrics, USA, 5 L min$^{-1}$) | $PM_{10}$ and $PM_{2.5}$ | 3.6% |
| | | | Yamamoto et al., 2012 | Eight-stage Andersen sampler (New Star Environmental, Roswell, GA, USA) | PM with aerodynamic diameter is 2.1-3.3, 3.3-4.7, 4.7-5.8, 5.8-9.0 and >9.0 μm | Abundant |

**Table 3.**

| Taxa | PM$_1$ | | | PM$_{2.5}$ | | | $p$ value | q value |
|---|---|---|---|---|---|---|---|---|
| | Mean | Std.err | Variance | Mean | Std.err | Variance | | |
| Glomerella | 10.51984 | 0.021813 | 0.013798 | 22.49025 | 0.01807 | 0.009796 | 0.000999 | 0.025543 |
| Zasmidium | 6.523201 | 0.011769 | 0.004017 | 12.71881 | 0.012239 | 0.004494 | 0.000999 | 0.025543 |
| Phyllosticta | 2.507228 | 0.004038 | 0.000473 | 5.948659 | 0.004366 | 0.000572 | 0.000999 | 0.025543 |
| Preussia | 0.039161 | 0.000195 | 1.10E-06 | 0.009109 | 6.81E-05 | 1.39E-07 | 0.0022322 | 0.042885 |
| Truncatella | 0.030579 | 0.00024 | 1.67E-06 | 0.005152 | 2.44E-05 | 1.79E-08 | 0.002784 | 0.046274 |
| Umbelopsis | 0.027549 | 0.000252 | 1.84E-06 | 0.005369 | 2.55E-05 | 1.95E-08 | 0.001669 | 0.034675 |
| Sebacina | 0.021306 | 0.000196 | 1.11E-06 | 0.001261 | 1.26E-05 | 4.77E-09 | 0.000550 | 0.022857 |
| Cordyceps | 0.020939 | 0.000137 | 5.45E-07 | 0.002518 | 1.75E-05 | 9.18E-09 | 0.001392 | 0.030848 |

**Figure 1.**

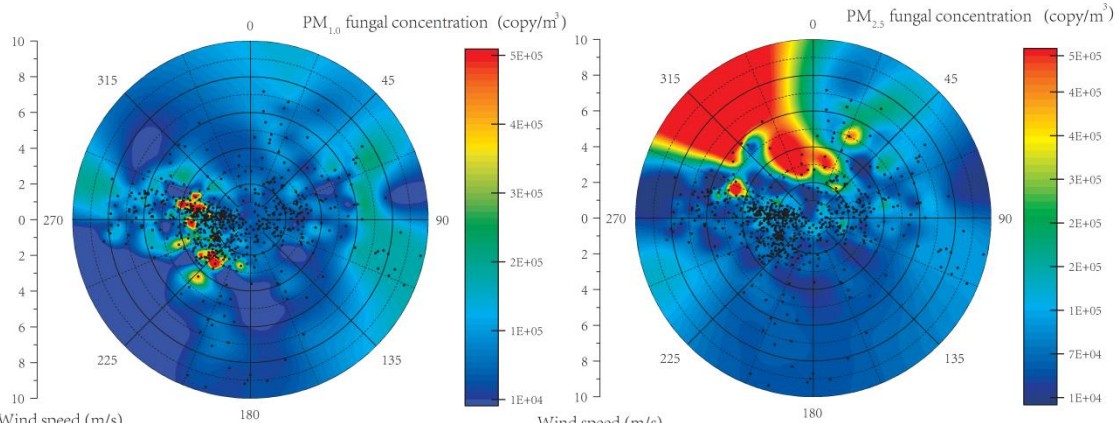

**Figure 2.**

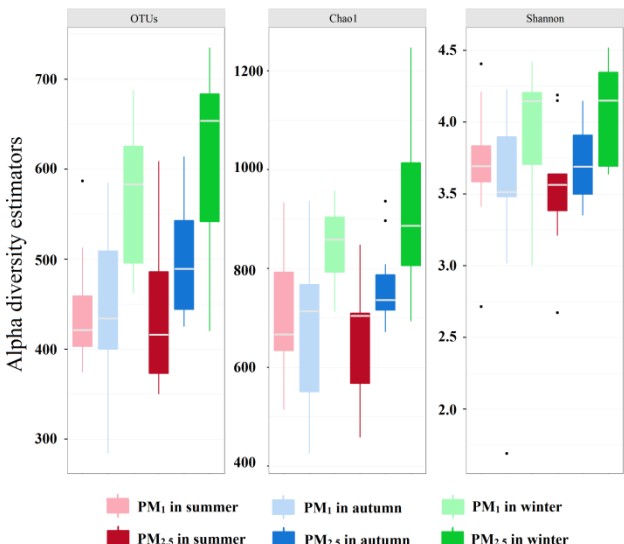

**Figure 3.**

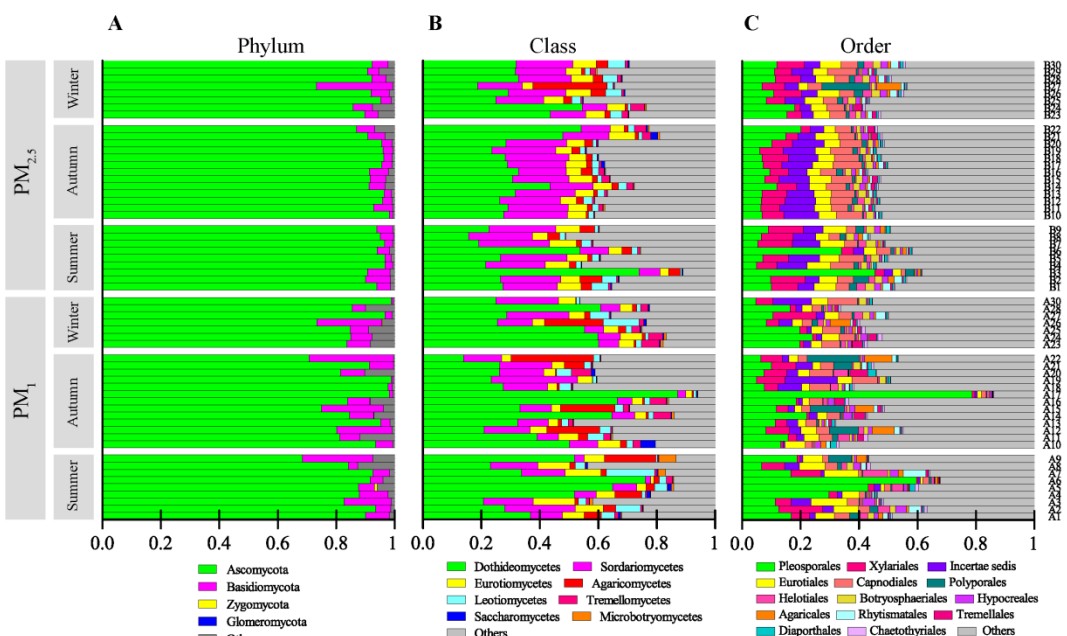

**Figure 4.**

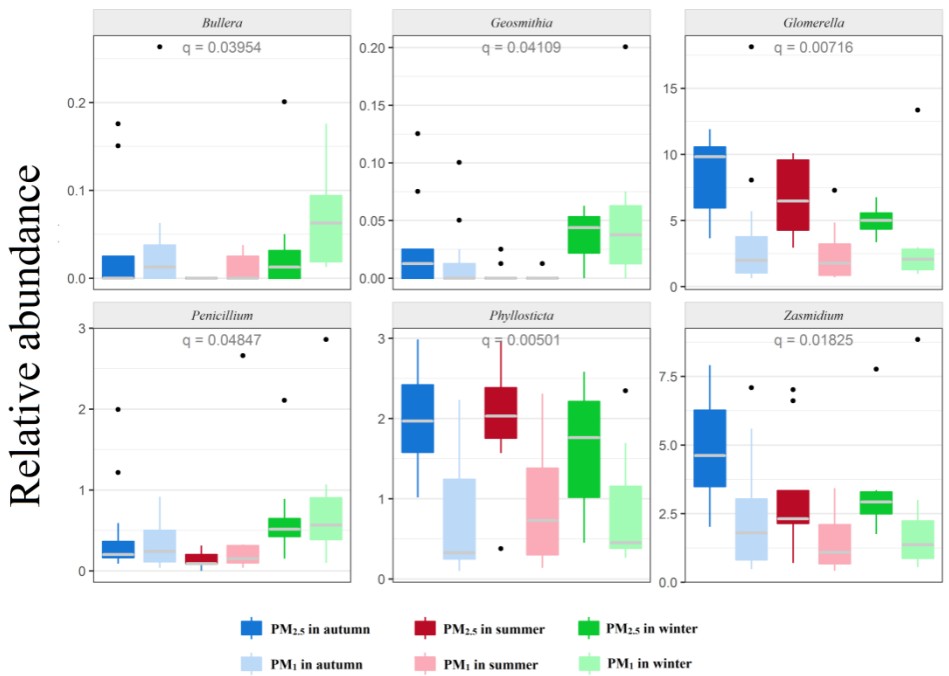

**Figure 5.**

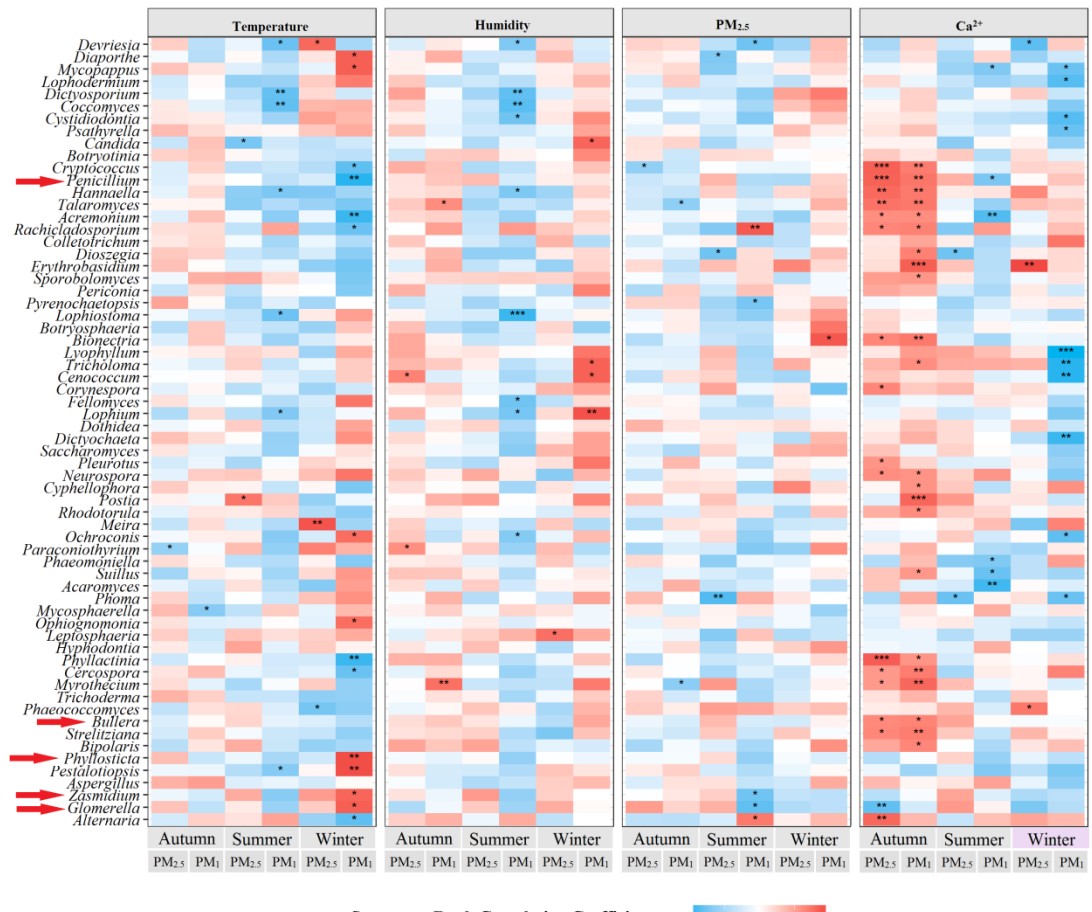

Spearman Rank Correlation Coefficient

-1  -0.5  0  0.5  1

