# Peer review of "Fungi Diversity in PM2.5 and PM1 at the summit of Mt. Tai: Abundance, Size Distribution, and Seasonal Variation"

_Atmospheric Chemistry and Physics, 2017_

## Referee Comment (RC1) · Anonymous Referee #1 · 30 Mar 2017

The study reported in this discussion paper describes fungal compositions and diversities in airborne PM1 and PM2.5 fractions that were collected at the summit of Mt. Tai, China. The study used quantitative PCR and high-throughput sequencing for the analyses of airborne fungal communities. I have several technical concerns, which are described as follows:

Major comments

Page 3 Line 21 More detailed information of the air samplers used in this study should be reported. Are they inertial impactors? If so, what is the sharpness of cutoff diameters for each stage? Also, how was particle bounces were prevented from the upper stages? Particle bounce can significantly distort the measured particle size distributions (e.g., Dzubay et al. (1976) Atmospheric Environment 10(3), 229-234). In particular, large particles can bounce from upper stages, because of their large inertia, and can penetrate through impactors and reach to an afterfilter even though they are not in fine fractions. If impactors were used, please state how particle bounces were prevented.

Page 4 Line 24 Were the chimeric sequences removed? The researchers reported more than 10% of the ITS sequences submitted to the public archives contained chimeric reads (Nilsson et al., (2015) Microbes and Environments 30(2): 145-150). This might affect the alpha diversity analyses, so it may be better to check.

Page 5 Line 5 How were airborne fungal concentrations calculated? Specifically, how did the investigators confirm or assume DNA extraction efficiency from fungal spores from air filters? It can affect final air concentrations reported.

Page 7 Line 39 Fungi in the class Dothideomycetes, including Alternaria, produce large multicellular spores, with reported spore sizes of $18–83 \times 7–18$ $\mu$m for Alternaria (Cole and Samson (1984) Mould allergy. Lea & Fibiger: Philadelphia, pp 66–104). It is hard to believe Alternaria was found in PM1 fraction, given with their large spore sizes, and I suspect it might be caused by sampling artifacts (e.g., particle bounces).

Minor comments

Page 6 Line 25 I could not understand why straw combustion can contribute airborne fungal DNA in PM1.

Page 7 Line 5 Do the authors believe 0.067% and 0.096% contributions truly non-negligible?

---

## Referee Comment (RC2) · Anonymous Referee #2 · 2 Apr 2017

The authors have studied the fungal diversity in PM1 and PM2.5 collected from M. Tai in China using gene sequence method. While some of the results are certainly useful, however the scientific questions they were addressing were not clear, or at least not focused. Its current form more or less looks like a technical report, with most figures developed from commercialized gene sequence method. In their work, it seems they addressed a variety of issues, e.g., health effects (fungal pathogens), fungal contents in PM1, seasonal effects, etc., but they did not have a clear scientific question to address. The reason why they have selected M. Tai as a sampling site, but not ground, was not discussed in details. It is hard to use their data to derive its impact on current understanding of the aerobiology, at least not from its current form.

[Figure]

In addition, they did not do the culturing for their PM samples which is simple. I believe that there will be more fungal spores in PM2.5 than PM1 since fungal species are in general bigger. They only detected sequence copies not the whole fungal spores. It would be much better if they could provide optical images of their detected fungal spores both for PM1 and PM2.5. For their sequence data, it seems they did not perform a robust statistical analysis. Gene sequence results could be very different sometimes if not in the same batch of experiments. How did they address the QC issues in their work? For the guideline values (800 CFU/m3), usually they refer to culturable bacterial CFU, while in their report they detected sequences. For fungal concentration levels, 800 CFU/m3 is a lot higher for most places. Last, some sentences were too verbal, e.g., "got" bigger. What does <typically «100) mean? Also it is "cultuirng" not cultured" method.

One suggestion to improve their paper is to try differentiate M. Tai from ground as a less human impact location (although there are also a lot of visitors). In this way, they might argue that what is fungal level and composition in less polluted higher atmosphere, and further derive potential conclusion about their presence and impact on climate or other things.

---

## Author Comment (AC1) · 14 Apr 2017

**Fungi Diversity in $PM_1$ and $PM_{2.5}$ at the summit of Mt. Tai:**

**Abundance, Size Distribution, and Seasonal Variation**

Caihong Xu[1], Min Wei[1], Jianmin Chen[1,2,3,*], Chao Zhu[1], Jiarong Li[1], Ganglin Lv[1], Xianmang Xu[1], Lulu Zheng[2], Guodong Sui[2], Weijun Li[1], Bing Chen[1], Wenxing Wang[1], Qingzhu Zhang[1], Aijun Ding[3], Abdelwahid Mellouki[1,4]

[1] Environment Research Institute, School of Environmental Science and Engineering, Shandong University, Ji'nan 250100, China

[2] Shanghai Key Laboratory of Atmospheric Particle Pollution and Prevention (LAP³), Fudan Tyndall Centre, Department of Environmental Science & Engineering, Fudan University, Shanghai 200433, China

[3] Institute for Climate and Global Change Research, School of Atmospheric Sciences, Nanjing University, Nanjing 210023, Jiangsu, China

[4] Institut de Combustion, Aérothermique, Réactivité et Environnement, CNRS, 45071 Orléans cedex 02, France

*Correspondence to* J. M. Chen (jmchen@sdu.edu.cn or jmchen@fudan.edu.cn)

Response to Reviewer 1

The study reported in this discussion paper describes fungal compositions and diversities in airborne $PM_1$ and $PM_{2.5}$ fractions that were collected at the summit of Mt. Tai, China. The study used quantitative PCR and high-throughput sequencing for the analyses of airborne fungal communities. I have several technical concerns, which are described as follows:

We thank the reviewer for the beneficial comments on our manuscript. We respond to the reviewer comments in detail below. The responses to reviewer are in red.

Major comments
Page 3 Line 21 More detailed information of the air samplers used in this study should be reported. Are they inertial impactors? If so, what is the sharpness of cutoff diameters for each stage? Also, how was particle bounces were prevented from the upper stages? Particle bounce can significantly distort the measured particle size distributions (e.g., Dzubay et al. (1976) Atmospheric Environment 10(3), 229-234). In particular, large particles can bounce from upper stages, because of their large inertia, and can penetrate through impactors and reach to an after filter even though they are not in fine fractions. If impactors were used, please state how particle bounces were prevented.

Response of the authors: The samplers used in this study were inertial impactors. Sampling were conducted by two middle volume air samplers (TH-100A, Wuhan Tianhong Instruments Co., Ltd., China), One of which was equipped with $PM_{2.5}$ fractionating inlet, the second one was equipped with a $PM_1$ fractionating inlet. Every sampler have only one stage, with the sharpness of cutoff diameters of them were 2.5 μm and 1 μm, respectively. Hence there are no issues for the particle bounces during the sampling periods. We have revises as in Page 3 in Line 21-25:

*Two middle volume inertial impactors (TH-100A, Wuhan Tianhong Instruments Co., Ltd., China, flow rate: 100 L min-1) equipped with $PM_{2.5}$ fractionating inlet and $PM_1$ fractionating inlet were used to collect $PM_{2.5}$ and $PM_1$, respectively. Sixty samples were obtained on the quartz membrane filters (PALL, NY, U.S., 88mm) for 23 h (9:00 am to 8:00 am next day) over 8-13 days during each season from 2014 to 2015 at the summit of Mt. Tai as shown in Table 1.*

Page 4 Line 24 Were the chimeric sequences removed? The researchers reported more than 10% of the ITS sequences submitted to the public archives contained chimeric reads (Nilsson et al., (2015) Microbes and Environments 30(2): 145-150). This might affect the alpha diversity analyses, so it may be better to check.

Response of the authors: The chimeric sequences were removed before alpha-diversity analysis. The sequences submitted to the public archives were the raw fastq files. But before the alpha diversity analysis and statistic analysis, we have trimmed the raw sequences by removing the chimeric sequences, low-quality sequences (length < 200 bp and Q value < 20), and chimera by FASTX-ToolKit (http://hannonlab.cshl.edu/fastx_toolkit) and Usearch (version 7.1 http://drive5.com/uparse/). The raw sequences and valid sequences were shown below.

Table 1 Raw sequences and valid sequences number of samples.

| No. | RS | VS | No. | RS | VS | No. | RS | VS |
|-----|-----|-----|-----|-----|-----|-----|-----|-----|
| A1 | 16770 | 14551 | A21 | 58617 | 51755 | B12 | 43968 | 39322 |
| A2 | 38089 | 32550 | A22 | 45199 | 40554 | B13 | 34925 | 31512 |
| A3 | 100967 | 79898 | A23 | 57862 | 46376 | B14 | 50917 | 44886 |
| A4 | 12236 | 9109 | A24 | 63683 | 50015 | B15 | 72251 | 63627 |
| A5 | 35098 | 20950 | A25 | 16412 | 13938 | B16 | 15817 | 13991 |
| A6 | 99119 | 51335 | A26 | 43746 | 38228 | B17 | 65677 | 57616 |
| A7 | 82450 | 66653 | A27 | 43877 | 38571 | B18 | 57527 | 52063 |
| A8 | 27325 | 24609 | A28 | 45251 | 38686 | B19 | 77755 | 70479 |
| A9 | 100807 | 47939 | A30 | 180380 | 164935 | B20 | 56931 | 48796 |
| A10 | 48298 | 44184 | B1 | 27627 | 24078 | B21 | 45951 | 38094 |
| A11 | 144435 | 137037 | B2 | 42178 | 38007 | B22 | 60784 | 50330 |
| A12 | 73806 | 65545 | B3 | 76494 | 53373 | B23 | 10202 | 8717 |

| A13 | 123617 | 111296 | B4 | 15338 | 13969 | B24 | 152770 | 127661 |
|-----|--------|--------|-----|-------|-------|-----|--------|--------|
| A14 | 47854 | 38137 | B5 | 56068 | 50431 | B25 | 48400 | 43593 |
| A15 | 38086 | 32625 | B6 | 13823 | 9797 | B26 | 47504 | 41459 |
| A16 | 100545 | 83655 | B7 | 70444 | 61531 | B27 | 63400 | 56821 |
| A17 | 25850 | 9008 | B8 | 58302 | 51779 | B28 | 50117 | 43514 |
| A18 | 35313 | 31841 | B9 | 17488 | 12427 | B29 | 81316 | 73897 |
| A19 | 61030 | 55692 | B10 | 41966 | 37598 | B30 | 34285 | 29926 |
| A20 | 21763 | 18760 | B11 | 29285 | 26147 | | | |

\* RS indicates Raw Sequences number

\* VS indicates Valid Sequences number

We have revised as in Page 4 in Line 32-37 and Page 5 in Line 6-7:

Page 4 in Line 32-37:

*After high-through sequencing, We removed the chimeric and low-quality sequences by FASTX-ToolKit (http://hannonlab.cshl.edu/fastx_toolkit) and UCHIME algorithm (Edge et al., 2011) before diversity analysis and statistic analysis. The remaining high quality sequences were normalized to 7973 reads in order to compare the different samples effectively and then clustered into Operational taxonomic units (OTUs) at 97% similarity cutoff using USEARCH software (version 7.1, http://drive5.com/uparse/).*

Page 5 in Line 6-7:

*The raw reads were deposited into the NCBI Sequence Read Archive (SRA) database under accession number SRR5146156.*

Page 5 Line 5 How were airborne fungal concentrations calculated? Specifically, how did the investigators confirm or assume DNA extraction efficiency from fungal spores from air filters? It can affect final air concentrations reported.

Response of the authors:

We use the real-time quantitative PCR for ITS region to calculate the gene copy numbers. The fungal concentration were estimated, assuming an average gene copy number of 200 per fungal genome (Lee et al., 2010, Science of the Total Environment 408:1349-1357).

To optimized the efficient of DNA extraction, We modified some steps of laboratory experiment (sample pretreatments, DNA extraction and purification) for the further high-throughput sequencing following Jiang (2015, Nature Protocols 10(5): 768-779). Half of filters (about 121.64cm$^2$) were cut into small pieces; inserted into 50 ml Falcon tubes filled with sterilized $1\times$PBS buffer; centrifuged at 200g for 3 h at 4$^o$C; collected the resuspension into a 0.2μm Supor 200 PES Membrane Disc Filter. We cut the PES Membrane Disc Filter into small pieces for DNA extraction. The filters were heated to 65$^o$C in PowerBead tubes for 15 min, vortexing for 15 min. The DNA were extracted according to the standard PowerSoil DNA isolation protocol and purified by AMPure XP bead purification. All the above steps

were carried out in a decontaminated biosafety cabinet. We have revised as in Page 4 in Line 5-13:

*To optimized the efficient of DNA extraction, We modified some steps of sample pretreatments and DNA extraction experiments for the sufficient DNA yields following Jiang et al., (2015). Half of filters (about 121.64 cm$^2$) were cut into small pieces, inserted into 50 ml Falcon tubes filled with sterilized 1×PBS buffer, and centrifuged at 200g for 3 h at 4$^o$C. The resuspension were collected into a 0.2μm Supor 200 PES Membrane Disc Filter. We cut the PES Membrane Disc Filter into small pieces. The filters were heated to 65$^o$C in PowerBead tubes for 15 min and then vortexing for 15 min. The DNA were extracted according to the standard PowerSoil DNA isolation protocol and purified by AMPure XP bead purification.*

Page 7 Line 39 Fungi in the class Dothideomycetes, including Alternaria, produce large multicellular spores, with reported spore sizes of 18-83μm – 7-18 μm for Alternaria (Cole and Samson (1984) Mould allergy. Lea & Fibiger: Philadelphia, pp 66–104). It is hard to believe Alternaria was found in PM1 fraction, given with their large spore sizes, and I suspect it might be caused by sampling artifacts (e.g., particle bounces).

Response of the authors: Among all the fungal genus detected, *Alternaria* were common in aerosol particles (e.g. TSP, $PM_{10}$, $PM_{5.8-9}$, $PM_{4.7-5.8}$, $PM_{3.3-4.7}$, and $PM_{2.5}$) (Yamamoto et al., 2012, The ISME Journal 6: 1801-181; Yan et al., 2016, Frontiers in Microbiology 7: 487; Hwang et al., 2016 Air Quality Atmospheric Health 9: 561-568). In $PM_1$, Huang (2016, Environmental Pollution 214: 202-210) also reported the existence of *Alternaria* using low-volume samplers (BGI, USA, 16.7 L min$^{-1}$) in Urumqi, China. In the present study, we collected the $PM_1$ and $PM_{2.5}$ with two middle-volume air samplers (TH-100A, Wuhan Tianhong Instruments Co., Ltd., China, 100 L min$^{-1}$). There were no particle bounces during sampling periods. While the existence of *Alternaria* may be influenced by the following reasons:

1) The ambient *Alternaria* are ageing and losing their activity following the air parcel movements. The inactive *Alternaria* were broken into small fragments with component size is equal to or lower than 1μm when suspending in the atmosphere. So we obtained the damaging fragments of *Alternaria* in $PM_1$.

2) Due to the relative higher rate of pump (100 L/min), some intact *Alternaria* were cut into small fragments by sharp $PM_1$ fractionating inlet before deposited into the quartz membrane filters.

Minor comments

Page 6 Line 25 I could not understand why straw combustion can contribute airborne fungal DNA in $PM_1$.

Response of the authors: Previously, Huffma (2010, Atmospheric Chemistry and Physics 10(7): 3215-3233) considered the fine fluorescent biological aerosol particle (<1μm) were influenced by long-range transport and anthropogenic sources such as combustion. At the sampling time of Autumn, no obvious straw combustion phenomenon were observed. And we monitored some long-range transported events based on the 48h backward trajectories of air parcels (Figure 1). The long-range transported air parcels mainly derived from outer Mongolia Areas (the famous dustiest place, Nov.6), Siberia regions (Nov 3rd and Nov. 12), and desert region (Nov.5). Influenced by the air movements from desert region, the corresponding fungal abundance in Nov.5 enriched from $6.18 \times 10^4$ to $103 \times 10^4$ copy m$^{-3}$ (about 16.7-folds). Similarly, the corresponding fungal abundance influenced by air parcels from Siberia regions enriched from $22.2 \times 10^4$ and $18.3 \times 10^4$ copy m$^{-3}$ (about 16.7-folds). So we hypothesized that the long-range transport of air parcels from north China may be the possible reason of the fungal enhancement of PM$_1$.

Figure 1 The 48h backward trajectory of air masses during Nov. 2014.

[Figure]

We have revised as in Page 7 Line 2-14:

*In this study, the highest fungal concentration in PM$_{2.5}$ was observed in summer (641 spores m$^{-3}$), whereas the highest value in PM$_1$ was found in autumn (1033 spores m$^{-3}$) indicated different origins of fungal spores. Huffman et al., (2010) found the long-range transport and anthropogenic sources such as combustion have a influence on the fluorescent biological aerosol particle with diameter smaller than 1μm. During sampling time in autumn, no obvious straw combustion phenomenon observed and we detected some long-range transported events in Nov. 2014. The long-range transported air parcels mainly derived from outer Mongolia*

*Areas (the famous dustiest place, Nov. 6), Siberia regions (Nov 3rd and Nov.12), and desert region (Nov. 5). Influenced by the air movements from desert region, the corresponding fungal abundance enriched from $6.18 \times 10^4$ to $103 \times 10^4$ copy $m^{-3}$ (about 16.7-folds). Similarly, the corresponding fungal abundance influenced by air parcels from Siberia regions enriched to $22.2 \times 10^4$ and $18.3 \times 10^4$ copy $m^{-3}$, respectively. So we hypothesized that the long-range transport of air parcels from north China may contributed to the fungal enhancement of $PM_1$.*

Page 7 Line 5 Do the authors believe 0.067% and 0.096% contributions truly nonnegligible ?

Response of the authors: Previously, the tracer-based methods were used for the fungal OC concentration based on micro-tracer levels such as polysaccharides, phospholipids, mannitol, proteins and ergosterol. Herein we calculated the fungal OC concentration according to the mannitol levels. The comparison with previous investigation used the same tracer were shown as below:

| Sampling Site | Sample Type | Fungal OC concentration | Fungal OC contribution to OC | Reference |
|---|---|---|---|---|
| Rinnbockstrasse, Austria | $PM_{10}$ | $0.3 \mu g\ m^{-3}$ | 8% | Bauer et al., 2008 |
| Schafberg, Austria | $PM_{10}$ | $0.35 \mu g\ m^{-3}$ | 14% | Bauer et al., 2008 |
| Beijing, China | $PM_{2.5}$ | $0.3\ \mu g\ C\ m^{-3}$ | 1.2% | Liang et al., 2017 |
| Beijing, China | $PM_{10}$ | $0.8\ \mu g\ C\ m^{-3}$ | 3.5% | Liang et al., 2017 |
| Hongkong, China | $PM_{2.5}$ | $3.7\ ng\ C\ m^{-3}$ | 0.1% | Cheng et al., 2009 |
| Hongkong, China | $PM_{10}$ | $9.7\ ng\ C\ m^{-3}$ | 0.2% | Cheng et al., 2009 |
| Mt.Tai, China | $PM_{2.5}$ | $6.1\ ng\ C\ m^{-3}$ | | In the present study |
| Mt.Tai, China | $PM_1$ | $8.3\ ng\ C\ m^{-3}$ | | In the present study |

Due to the limited filters for estimation of OC mass concentration in the present study, we compare the fungal OC contribution to the total PM concentration. Assuming that the OC contribution to $PM_{2.5}$ were 17.5% in Beijing (Wang et al., 2015 Environmental Monitoring and Assessment 187(3):143) and 17% in Hongkong (Ho et al. 2004 Atmospheric Environment 38(37): 6327-6335), the percentage of fungal OC to total $PM_{2.5}$ in Beijing (0.069%) and Hongkong (0.00588%) were compared with the value in this study (0.067%). Hence I think the remarkably precise number (0.067% and 0.096%) was believable. Although the fungal spores contributed a minor OC source of $PM_{2.5}$ and $PM_1$, but it is also can not be ignored. The trace fungal OC were also able to participated in the atmospheric process or involved in the human health. We have revised as in Page7 Line 30-33:

*The daily averaged concentrations of fungal OC in $PM_{2.5}$ and $PM_1$ were 6.1 and 8.3 ng C $m^{-3}$ with the corresponding contribution to PM were 0.067% and 0.096%, indicating that*

*airborne fungal spores acted as a minor source of carbonaceous aerosol can not be ignored at Mt. Tai.*

---

## Author Comment (AC2) · 14 Apr 2017

**Fungi Diversity in $PM_1$ and $PM_{2.5}$ at the summit of Mt. Tai: Abundance, Size Distribution, and Seasonal Variation**

Caihong Xu[1], Min Wei[1], Jianmin Chen[1,2,3,*], Chao Zhu[1], Jiarong Li[1], Ganglin Lv[1], Xianmang Xu[1], Lulu Zheng[2], Guodong Sui[2], Weijun Li[1], Bing Chen[1], Wenxing Wang[1], Qingzhu Zhang[1], Aijun Ding[3], Abdelwahid Mellouki[1,4]

[1] Environment Research Institute, School of Environmental Science and Engineering, Shandong University, Ji'nan 250100, China

[2] Shanghai Key Laboratory of Atmospheric Particle Pollution and Prevention (LAP[3]), Fudan Tyndall Centre, Department of Environmental Science & Engineering, Fudan University, Shanghai 200433, China

[3] Institute for Climate and Global Change Research, School of Atmospheric Sciences, Nanjing University, Nanjing 210023, Jiangsu, China

[4] Institut de Combustion, Aérothermique, Réactivité et Environnement, CNRS, 45071 Orléans cedex 02, France

*Correspondence to* J. M. Chen (jmchen@sdu.edu.cn or jmchen@fudan.edu.cn)

Response to Reviewer 2

The authors have studied the fungal diversity in $PM_1$ and $PM_{2.5}$ collected from Mt. Tai in China using gene sequence method. While some of the results are certainly useful, however the scientific questions they were addressing were not clear, or at least not focused. Its current form more or less looks like a technical report, with most figures developed from commercialized gene sequence method.

Response: We thank the reviewer for the beneficial comments on our manuscript. We have added the description about the scientific questions and redrawn the Figure 2, Figure 3, Figure 4 , and Figure 5 in the revised essay. We respond to the reviewer comments in detail below. The responses to reviewer are in red.

1. In their work, it seems they addressed a variety of issues, e.g., health effects (fungal pathogens), fungal contents in $PM_1$, seasonal effects, etc., but they did not have a clear scientific question to address.

Response of the authors: We have modified the scientific question in introduction as in Page 3

Line 12-16:

*The objectives of the present study were: (i) to fill the knowledge gaps regarding the information on ambient fungi of $PM_{2.5}$ and $PM_1$ from a high-elevation site over East Asia, (ii) to elucidate the size-resolved differences between the data of ambient fungal concentration, viable fungal community structure in different levels across different seasons, (iii) to estimate whether the environmental factors play a role in the variation of fungal characteristics at Mt. Tai.*

2. The reason why they have selected Mt. Tai as a sampling site, but not ground, was not discussed in details. It is hard to use their data to derive its impact on current understanding of the aerobiology, at least not from its current form.

Response of the authors: Thanks for your suggestion. We have added the description about the reason why we selected Mt. Tai as in Page 2 Line 34- Page 3 Line11:

*However, the relevant study commonly focused on the fungal communities in total suspended particles (TSP), $PM_{10}$, and $PM_{2.5}$ and primarily conducted over the ground surface, the attention on the fungal population in $PM_1$ at a high-elevation site were limited. Specific microbes at high altitudes (such as clouds water and precipitation) can act as nuclei and ice crystals and influence the precipitation patterns (Pratt et al., 2009; Creamean et al., 2013; Bower et al., 2013). Hence, it is essential to advance the knowledge, especially across East Asian region. During 2013, 2014 and 2015, serious air pollution events associated with the inadequate use of energy in the transport, domestic, and industrial sectors attacked the Northern China, which is the seriously air polluted areas including Beijing, Tianjin, Shijiazhuang, Jinan, and Qingdao. Mt. Tai (36°15'N, 117°06'E, 1534 m a.s.l), the highest site in the North China Plain, is a tilted fault block mountain with height increasing from the north to the south, facing to the Japanese Islands, Korean Peninsula, East China Sea, and Yellow Sea. The vegetation coverage reach to 80% and there are nearly 1000 kinds of plants grow in this area. In 2014 and 2015, the number of tourists from both China and abroad has increased from 5.5 to 5.9 million. In this area, the previous investigations mainly concentrated on the physicochemical characteristics of aerosol particles and cloud water and their influence on the air quality and human health. So far there were no researches focused on the diverse fungal community in aerosol particles at Mt. Tai. It is necessary to build a sophisticated finished knowledge on the atmospheric aerosol in such scenic outlook.*

3. In addition, they did not do the culturing for their PM samples which is simple. I believe that there will be more fungal spores in $PM_{2.5}$ than $PM_1$ since fungal species are in general bigger. They only detected sequence copies not the whole fungal spores. It would be much better if they could provide optical images of their detected fungal spores both for $PM_1$ and $PM_{2.5}$.

Response of the authors: In the present study, samples were collected into 88mm quartz membrane filters. Half of filter was cut for DNA extraction for fungal concentration and community, and the remaining was used for the analysis of water soluble inorganic ions. So It is too pity. We have not enough filters and mature technology for the culturing and optical images. In the future, we will collect more filters for theses two analysis.

4. For their sequence data, it seems they did not perform a robust statistical analysis. Gene sequence results could be very different sometimes if not in the same batch of experiments. How did they address the QC issues in their work?

Response of the authors: Thanks for your suggestion. Though the sampling experiment lasted almost two years (May 2014-Aug. 2015), all samples were stored at $-80^{o}C$ till the DNA extraction. We selected sixty representative samples (A1-A30, B1-B30) when the field measurements finished. I am assured that the laboratory experiments of $PM_{2.5}$ and $PM_1$ were conducted in a same batch of experiments including DNA extraction, PCR amplication, real-time qPCR, and Illumina Sequencing except A29 (accidentally omitted in the first batch of Illumina Sequencing). Considering the fact that sequence varied different in different batches of experiments, we remove the A29 before quality control. A robust statistical analysis of raw sequences were preformed before diversity and taxonomic analysis. After Miseq sequencing, the raw sequences were saved by Fastq files. The Q value (Phred quality score) were calculated by the following equation:

$$Qphred=-10\log_{10}(p)$$

*p indicates the base read error rate

The paired reads were jointed together into sequences by soft FLAST. The quality control were conducted includes: a) removing the primers and barcodes; b) removing the low-quality sequences (length < 250 bp and Q value < 20); c) removing the chimeric sequences. The valid sequences were shown as below.

Table 1 Raw sequences and valid sequences number of samples.

| No. | RS | VS | No. | RS | VS | No. | RS | VS |
|-----|----|----|-----|----|----|-----|----|----|
| A1 | 16770 | 14551 | A21 | 58617 | 51755 | B12 | 43968 | 39322 |
| A2 | 38089 | 32550 | A22 | 45199 | 40554 | B13 | 34925 | 31512 |
| A3 | 100967 | 79898 | A23 | 57862 | 46376 | B14 | 50917 | 44886 |
| A4 | 12236 | 9109 | A24 | 63683 | 50015 | B15 | 72251 | 63627 |
| A5 | 35098 | 20950 | A25 | 16412 | 13938 | B16 | 15817 | 13991 |
| A6 | 99119 | 51335 | A26 | 43746 | 38228 | B17 | 65677 | 57616 |
| A7 | 82450 | 66653 | A27 | 43877 | 38571 | B18 | 57527 | 52063 |
| A8 | 27325 | 24609 | A28 | 45251 | 38686 | B19 | 77755 | 70479 |
| A9 | 100807 | 47939 | A30 | 180380 | 164935 | B20 | 56931 | 48796 |
| A10 | 48298 | 44184 | B1 | 27627 | 24078 | B21 | 45951 | 38094 |

| | | | | | | | | |
|---|---|---|---|---|---|---|---|---|
| A11 | 144435 | 137037 | B2 | 42178 | 38007 | B22 | 60784 | 50330 |
| A12 | 73806 | 65545 | B3 | 76494 | 53373 | B23 | 10202 | 8717 |
| A13 | 123617 | 111296 | B4 | 15338 | 13969 | B24 | 152770 | 127661 |
| A14 | 47854 | 38137 | B5 | 56068 | 50431 | B25 | 48400 | 43593 |
| A15 | 38086 | 32625 | B6 | 13823 | 9797 | B26 | 47504 | 41459 |
| A16 | 100545 | 83655 | B7 | 70444 | 61531 | B27 | 63400 | 56821 |
| A17 | 25850 | 9008 | B8 | 58302 | 51779 | B28 | 50117 | 43514 |
| A18 | 35313 | 31841 | B9 | 17488 | 12427 | B29 | 81316 | 73897 |
| A19 | 61030 | 55692 | B10 | 41966 | 37598 | B30 | 34285 | 29926 |
| A20 | 21763 | 18760 | B11 | 29285 | 26147 | | | |

\* RS indicates Raw Sequences number

\* VS indicates Valid Sequences number

We have revised as in Page 3 in Line 34-38 and Page 4 Line 32-37:

Page 3 in Line 34-38:

*The remaining filters were analyzed in a same batch of lab experiments including DNA extraction, PCR amplication, real-time qPCR, and Illumina Sequencing except the sample A29 in Dec. 9, 2014 (accidentally omitted in the first batch of Illumina Sequencing). Considering the fact that a percentage of sequences in two batches of experiments were different, we removed this sample before quality control.*

Page 4 Line 32-37:

*After high-through sequencing, We removed the chimeric and low-quality sequences by FASTX-ToolKit (http://hannonlab.cshl.edu/fastx_toolkit) and UCHIME algorithm (Edge et al., 2011) before diversity analysis and statistic analysis. The remaining high quality sequences were normalized to 7973 reads in order to compare the different samples effectively and then clustered into Operational taxonomic units (OTUs) at 97% similarity cutoff using USEARCH software (version 7.1, http://drive5.com/uparse/).*

5. For the guideline values (800 CFU/m3), usually they refer to culturable bacterial CFU, while in their report they detected sequences. For fungal concentration levels, 800 CFU/m3 is a lot higher for most places.

Response of the authors: Thanks for your suggestion. The guideline (800 CFU $m^{-3}$) was developed for the culturable fungal CFU by Chinese Academy of Sciences Ecological Environmental Research Center. To date, there is no uniform guidelines for the fungal concentration based on the qPCR. We have deleted the unreasonable comparison with this guideline value (800 CFU $m^{-3}$).

6. Last, some sentences were too verbal, e.g., "got" bigger. What does <typically «l00) mean? Also it is "cultuirng" not cultured" method.

Response of the authors: Thanks for your suggestion, the expression "typically << 100" means "typically less than 100" and " got bigger" means "the particle size increased". I have revised as in Page 2 in Line 19 and Page 1 in Line 27. The remaining verbal sentences were modified by a native English speaker and highlighted in the revised manuscripts.

7. One suggestion to improve their paper is to try differentiate Mt. Tai from ground as a less human impact location (although there are also a lot of visitors). In this way, they might argue that what is fungal level and composition in less polluted higher atmosphere, and further derive potential conclusion about their presence and impact on climate or other things.

Response of the authors: Thanks for your suggestion, we have revised largely the introduction and discussion sections in the revised manuscript.

---

## Referee Comment (RC3) · Anonymous Referee #1 · 15 Apr 2017

The authors did not really address my point about particle bounce. The authors insisted no particle bounce, but they did not provide evidences or reasons of why they can insist so. What measures were taken to prevent particle bounce from the fractionating inlets? I assume the fractionating inlets (i.e., impactors) remove all particles larger than 1 or 2.5 $\mu$m, and the remaining fractions (i.e., PM1 and PM2.5) were collected on afterfilters. This point is very important since the study is intended to report fungal communities in PM1 and PM2.5 fractions. It is possible the authors merely measured accumulations of bounced particles from the fractionating inlets that were not really of PM1 and PM2.5 portions.

The authors seems to misunderstand the definition of sharpness of cutoff diameters by

impactors. Cutoff diameter and sharpness of impactors are different. See, for example, Huang, J Air Waste Manag Assoc. 2005;55(12):1858-65 for a definition of sharpness of cutoff diameters.

The details of DNA extraction protocol were provided, but information about extraction efficiency was not provided. If DNA extraction efficiency is unknown or un-assumed, DNA concentrations in air cannot be back-calculated.

The authors explained two possible reasons of why Alternaria can be found in PM1 and PM2.5 fractions. The second explanation of fragmentation by the sampler's inlets is problematic. If it is so, the sizes of Alternaria reported in this study were not really representative of their sizes in air. I assume the purpose of this study was to report their sizes in air, not the sizes of spores fragmented by the sampler's inlets.

---

## Author Comment (AC3) · 17 May 2017

**Fungi Diversity in $PM_1$ and $PM_{2.5}$ at the summit of Mt. Tai: Abundance, Size Distribution, and Seasonal Variation**

Response to Reviewer 1

We thank the reviewer for the beneficial comments on our manuscript. We respond to the reviewer comments in detail below. The responses to reviewer are in red. The abundant fungal genus and top five orders in our study were listed in the following table. They were also widely found in the suspending particles including TSP, $PM_{10}$, $PM_{2.5}$, and $PM_1$. The results we obtained were reasonable and effective.

| Common Fungi | RAS[a] | RAF[b] | References | Samplers | Sample Type | Concentration or Abundance |
|---|---|---|---|---|---|---|
| Alternaria | 11.7 | 6.2 | Hameed et al., 2012 | Slit impactor sampler (Model П818N°5587, САЕПАНО, ВСССР) | TSP | 26.5 CFU/m³ |
| | | | Adhikari et al., 2004 | Andersen sampler (Thermo Andersen, Smyrna, 300082-5211, USA) | TSP | 2.6% |
| | | | Oh et al., 2014 | High volume air sampler (Model no.5000; E &Instrument, Korea) | TSP | Abundant genera |
| | | | Shelton et al., 2002 | Andersen N6 samplers (Thermo Andersen, Inc., Atlanta, Ga.) | TSP | |
| | | | Hwang et al., 2016 | 17G9 GilAir Sampler (Gilian Product Sensidyne, Inc., USA) | TSP | |
| | | | Dannemiller et al., 2014 | High volume PM10 samplers (Ecotech, Knoxfield, VIC, Australia) | $PM_{10}$ | >1% |
| | | | Alghamdi et al., 2014 | $PM_{2.5}$ samplers (Staplex Air Sampler Division, USA) | $PM_{2.5}$ | 2.6% |
| | | | Gou et al., 2016 | Low volume air sampler (BGI, USA) | $PM_{10}$ and $PM_1$ | >1% |
| Aspergillus | 2.3 | 1.9 | Hameed et al., 2012 | Slit impactor sampler (Model П818N°5587, САЕПАНО, ВСССР) | TSP | 103.98 CFU/m³ |
| | | | Alghamdi et al., 2014 | $PM_{10}$ samplers (Staplex Air Sampler Division, USA) | $PM_{10}$ | 13.1 CFU/m³ |
| | | | Alghamdi et al., 2014 | $PM_{2.5}$ samplers (Staplex Air Sampler Division, USA) | $PM_{2.5}$ | 7.9 CFU/m³ |
| | | | Cao et al., 2014 | Air samplers (Thermo Electron Corp., MA, U.S.) | $PM_{10}$ and $PM_{2.5}$ | Abundant genera |
| | | | Gou et al., 2016 | Low volume air sampler (BGI, USA) | $PM_{10}$ and $PM_1$ | Abundant genera |
| Pleosporales | 18.4 | 45.4 | Rittenour et al., 2014 | Buck Bioaire Sampler (A.P. Buck, Inc, Orlando, FL, USA) | TSP | 46% |
| | | | Yan et al., 2016 | Air samplers (Air Metrics, USA, 5 L/min) | $PM_{10}$ and $PM_{2.5}$ | 29.4% |
| | | | Gou et al., 2016 | Low volume air sampler (BGI,USA) | $PM_{10}$ and $PM_1$ | 10-15% |

| | RAS[a] | RAF[b] | Reference | Sampler | PM type | Relative abundance |
|---|---|---|---|---|---|---|
| Xylariales | 5.0 | 14.4 | Womack et al., 2015 | SKC Biosamplers (BioSampler SKC Inc.) | TSP | Abundant order |
| | | | Gou et al., 2016 | Low volume air sampler (BGI, USA) | $PM_{10}$ and $PM_1$ | 0-5% |
| Eurotiales | 4.8 | 13.3 | Yan et al., 2016 | Air samplers (Air Metrics, USA, 5 L/min) | $PM_{10}$ and $PM_{2.5}$ | 10.6% |
| | | | Gou et al., 2016 | Low volume air sampler (BGI, USA) | $PM_{10}$ and $PM_1$ | 10-15% |
| Capnodiales | 4.4 | 12.5 | Yan et al., 2016 | Air samplers (Air Metrics, USA, 5 L/min) | $PM_{10}$ and $PM_{2.5}$ | 27.96% |
| | | | Gou et al., 2016 | Low volume air sampler (BGI, USA) | $PM_{10}$ and $PM_1$ | ~25% |
| Polyporales | 2.5 | 6.4 | Womack et al., 2015 | SKC Biosamplers (BioSampler SKC Inc.) | TSP | Abundant order |
| | | | Yan et al., 2016 | Air samplers (Air Metrics, USA, 5 L/min) | $PM_{10}$ and $PM_{2.5}$ | 3.6% |
| | | | Yamamoto et al., 2012 | Eight-stage Andersen sampler (New Star Environmental, Roswell, GA, USA) | PM with aerodynamic diameter is 2.1-3.3, 3.3-4.7, 4.7-5.8, 5.8-9.0 and >9.0 μm | Abundant order |

RAS[a] indicates Relative Abundance in Submicron particles.
RAF[b] indicates Relative Abundance in Fine particles.

1. The authors did not really address my point about particle bounce. The authors insisted no particle bounce, but they did not provide evidences or reasons of why they can insist so. What measures were taken to prevent particle bounce from the fractionating inlets? I assume the fractionating inlets (i.e., impactors) remove all particles larger than 1 or 2.5 μm, and the remaining fractions (i.e., $PM_1$ and $PM_{2.5}$) were collected on after filters. This point is very important since the study is intended to report fungal communities in $PM_1$ and $PM_{2.5}$ fractions. It is possible the authors merely measured accumulations of bounced particles from the fractionating inlets that were not really of $PM_1$ and $PM_{2.5}$ portions.

Response of the authors:

The samplers we used were commercial instruments and the design and quality control were qualified. Actually the particle bounce phenomenon existed in the inertial samplers. To prevent particle bounce from the fractionating inlets, we smear some silicone oil over the inside of drip catcher in every stage (as shown below) before sampling. And the samplers were operated strictly according to the manufactures' direction. We assured that the collected particles were $PM_{2.5}$ and $PM_1$ rather than the accumulation of bounced particles.

[Figure]

38-, 26-,12-, 6-well strike plate

Drip Catcher

2. The authors seems to misunderstand the definition of sharpness of cutoff diameters by impactors. Cutoff diameter and sharpness of impactors are different. See, for example, Huang, J Air Waste Manag Assoc. 2005;55(12):1858-65 for a definition of sharpness of cutoff diameters.

Response of the authors:

The cutoff aerodynamic diameter(da50) is defined as the aerodynamic diameters of particle when the collection efficiency was 50%.

The sharpness (GSD) were defined as follows: GSD=$\sqrt{\frac{da84\%}{da16\%}}$.

*da84% means the aerodynamic diameters of particle at 84% collection efficiency.

*da16% means the aerodynamic diameters of particle at 16% collection efficiency.

For the PM$_{2.5}$ sampler, the cutoff and sharpness diameter were 2.5μm and 0.80, respectively.

For the PM$_1$ sampler, the cutoff and sharpness diameter were 1μm and 0.71, respectively.

We have revised as in Page Line:

*Two middle volume inertial impactors (TH-150, Wuhan Tianhong Instruments Co., Ltd., China, 100 L min-1), corresponding to the cut-off diameter of 2.5μm and 1μm, were employed to collect PM$_{2.5}$ and PM$_1$ samples, respectively. We obtained sixty quartz membrane filters (PALL, NY, USA, 88mm) for 23 h (9:00 am to 8:00 am next day) over 8-13 days during each season from 2014 to 2015 at the summit of Mt. Tai as shown in Table 1.*

3. The details of DNA extraction protocol were provided, but information about extraction efficiency was not provided. If DNA extraction efficiency is

unknown or un-assumed, DNA concentrations in air cannot be back-calculated.

Response of the authors:
Generally, the recover efficiency should be 60%-80%. The value depends on the type of sample. Some turbid samples might come lower efficiency as it is a bit hard to lyse, but for water it will be higher. In the lab experiment, I have added some external control in exact copy number to the sample firstly, and check with the copy number after extraction. The efficiency was 71.29%. Below is the data for the expected yield and specification of the kit.

| Sample | Relative Quantity | Relative Quantity SD | Corrected Relative Quantity SD | Relative Quantity SEM | Corrected Relative Quantity SEM | Mean Cq | Cq SD | Cq SEM | Ln (Copy Number) | Copy Number | Copies per μl |
|---|---|---|---|---|---|---|---|---|---|---|---|
| Pre OP | 0.4191 | 0.0730 | 0.0739 | 0.0516 | 0.0522 | 17.4340 | 0.2462 | 0.1741 | 13.3790 | 646304.6385 | 64630463.8462 |
| Post OP | 0.5862 | 0.0062 | 0.0115 | 0.0044 | 0.0081 | 16.9597 | 0.0149 | 0.0105 | 13.7174 | 906567.4378 | 90656743.7794 |
| | | | | | | | | | | Efficiency | 71.29% |

4. The authors explained two possible reasons of why Alternaria can be found in $PM_1$ and $PM_{2.5}$ fractions. The second explanation of fragmentation by the sampler's inlets is problematic. If it is so, the sizes of Alternaria reported in this study were not really representative of their sizes in air. I assume the purpose of this study was to report their sizes in air, not the sizes of spores fragmented by the sampler's inlets.

Response of the authors:
Based on the guideline developed by the China's Ministry of Environmental Protection (HJ93-2013), the design of samplers were eligible and effective. The $PM_1$ sampler was composed of the $PM_{2.5}$ sampler and one more stage (6-well 2.5mm strike plate and drip catcher) before the filter. For $PM_1$ sampler, the da16% is 1.18 μm and da84% is 0.83 μm, respectively. The geometric standard deviation of sampling efficiency (σg) is 1.2±0.1. The fragment of fungal spores caused by the sample's inlets were existed indeed. But this phenomenon was tiny and within a reasonable range. We operated the samplers strictly according to the manufactures' directions and have applied this instruments to the researches on the size distribution of aerosol particles (Zhang et al., 2016; Zhao et al., 2017).

[Figure]

**For PM₂.₅**

38-well Strike plate, 5.1 mm

26-well Strike plate, 4.1 mm

12-well Strike plate, 3.2 mm

**For PM₁**

38-well Strike plate, 5.1 mm

26-well Strike plate, 4.1 mm

12-well Strike plate, 3.2 mm

6-well Strike plate, 2.5 mm

**References**

Hameed A A A, Khoder M I, Ibrahim Y H, et al. Study on some factors affecting survivability of airborne fungi. Science of the Total Environment, 2012, 414(1): 696-700.

Adhikari A, Sen M M, Gupta-Bhattacharya S, et al. Airborne viable, non-viable, and allergenic fungi in a rural agricultural area of India: a 2-year study at five outdoor sampling stations. Science of the Total Environment, 2004, 326(1–3):123-141.

Shelton B G, Kirkland K H, Flanders W D, et al. Profiles of Airborne Fungi in Buildings and Outdoor Environments in the United States[J]. Applied & Environmental Microbiology, 2002, 68(4):1743.

Hwang S H, Cho J H. Evaluation of airborne fungi and the effects of a platform screen door and station depth in 25 underground subway stations in Seoul, South Korea. Air Quality, Atmosphere & Health, 2016, 9(5):561-568.

Dannemiller KC., Lang-Yon N, Yamamoto N, Rudich Y, Peccia J. Combining real-time PCR and next-generation DNA sequencing to provide quantitative comparisons of fungal aerosol populations. Atmospheric Environment, 2014, 84(84):113–121.

Almaguer M, Aira M J, Rodríguez-Rajo F J, et al. Temporal dynamics of airborne fungi in Havana (Cuba) during dry and rainy seasons: influence of meteorological parameters. International Journal of  Biometeorology, 2014, 58(7):1459-1470.

Gou H, Lu J, Li S, et al. Assessment of microbial communities in PM1, and PM10, of Urumqi during winter . Environmental Pollution, 2016, 214:202.

Cao C, Jiang W, Wang B, et al. Inhalable Microorganisms in Beijing's PM2.5 and PM10 Pollutants during a Severe Smog Event. Environmental Science & Technology, 2014, 48(3):1499.

Rittenour W R, Ciaccio C E, Barnes C S, et al. Internal transcribed spacer rRNA gene sequencing analysis of fungal diversity in Kansas City indoor environments. Environmental Science Processes & Impacts, 2013, 16(1): 33-43.

Yan D, Zhang T, Su J, et al. Diversity and Composition of Airborne Fungal Community Associated with Particulate Matters in Beijing during Haze and Non-haze Days. Frontiers in Microbiology, 2016, 7:487.

Yamamoto N, Bibby K, Qian J, et al. Particle-size distributions and seasonal diversity of allergenic and pathogenic fungi in outdoor air. ISME Journal, 2012, 6(10):1801.

Womack A M, Artaxo P E, Ishida F Y, et al. Characterization of active and total fungal communities in the atmosphere over the Amazon rainforest. Biogeosciences Discussions, 2015, 12(10):1176-1176.

Huang C H. Predicting cutoff aerodynamic diameter and sharpness of single round-nozzle impactors with a finite impaction plate diameter. Journal of the Air & Waste Management Association, 2005, 55(12):1858-65.

Zhang J, Yang L, Chen J, et al. Influence of fireworks displays on the chemical characteristics of PM 2.5, in rural and suburban areas in Central and East China. Science of the Total Environment, 2016.

Zhao T, Yang L, Yan W, et al. Chemical characteristics of PM1/PM2.5 and influence on visual range at the summit of Mount Tai, North China. Science of the Total Environment, 2017, 575:458-466.

---

## Referee Comment (RC4) · Anonymous Referee #1 · 18 May 2017

The reviewer appreciate the responses made by the authors. As the investigators did, impactors have to be coated by oil or grease to prevent from bounce of giant particles.

The reviewer appreciate the additional experiment done by the authors about DNA extraction efficiency. However, the efficiency reported in the response seems to be recovery efficiency of DNA associated purification and elution steps, and not efficiency of DNA extraction from fungal cells. In this case, the authors had to spike fungal spores, not naked fungal DNA or PCR amplicons, as an external control. Usually, 70

The reviewer is also curious about the concentrations reported in the abstract. The authors reported that the fungal abundance was $9.4 \times 104$ and $1.3 \times 105$ copies m-3 in PM2.5 and PM1, respectively. However, this is against basic law of physics. The

concentration has to be higher in PM2.5 than in PM1 because PM2.5 is inclusive of PM1.

---

## Author Response (AR1)

**Response to the Referees**

We thank the reviewer for the beneficial comments on our manuscript. We have revised the manuscript largely according to your comments. The verbal sentences have been modified by a native English speaker and highlighted in the revised manuscript. We respond to the reviewer comments in detail below. The responses to reviewer are in red.

**Comments from Anonymous Referee #1**

1, The reviewer appreciate the responses made by the authors. As the investigators did, impactors have to be coated by oil or grease to prevent from bounce of giant particles. The reviewer appreciate the additional experiment done by the authors about DNA extraction efficiency. However, the efficiency reported in the response seems to be recovery efficiency of DNA associated purification and elution steps, and not efficiency of DNA extraction from fungal cells. In this case, the authors had to spike fungal spores, not naked fungal DNA or PCR amplicons, as an external control.

Response of the authors: Thanks for your suggestions. Compared to the plate count method, direct microscopic examination, spectrographic method, and biosensor method, the real-time qPCR has higher specificity, simplicity, and convenience in operation. This method can provides quantitative DNA or RNA data and have be readily applicable to the detection of airborne microorganisms in different environmental samples (Gao et al., 2017; Yamaguchi et al., 2016; Gandolfi et al., 2015; DeLeon-Rodriguez et al., 2013; Lee et al. 2010; Lindsley et al., 2010; Anne et al., 2008; Alexander et al., 2006).

The DNA extraction efficiency is the ratio of extracted DNA concentration to the original DNA concentration. If we use the fungal spores, I am confused about how to determine the original amount of DNA from fungal spores. Could you please provide the standard operation protocol of this experiment?

2, The reviewer is also curious about the concentrations reported in the abstract. The authors reported that the fungal abundance was $9.4 \times 10^4$ and $1.3 \times 10^5$ copies m-3 in $PM_{2.5}$ and $PM_1$, respectively. However, this is against basic law of physics. The concentration has to be higher in $PM_{2.5}$ than in $PM_1$ because $PM_{2.5}$ is inclusive of $PM_1$.

Response of the authors: Conceptually, $PM_{2.5}$ and $PM_1$ were defined as the particles with aerodynamic equivalent diameter less than or equal to 2.5μm and 1μm. But in reality, the size of aerosol particle we captured were 2.5±0.2μm and 1±0.2μm when the collect efficiency

varied from 16% to 84%. In this paper, $PM_1$ was not included in the $PM_{2.5}$ and thus I think there is no direct affiliation between these two type particles.

| Type | Defined particle size | Practical particle size in this study |
|------|----------------------|----------------------------------------|
| $PM_{2.5}$ | $\leq 2.5\mu m$ | $= 2.5\pm0.2\mu m$ |
| $PM_1$ | $\leq 1\mu m$ | $= 1\pm0.2\mu m$ |

Response of the authors: Thanks for your suggestion. Though the sampling experiment lasted almost two years (May 2014-Aug. 2015), all samples were stored at -80$^{\circ}$C till the DNA extraction. We selected sixty representative samples (A1-A30, B1-B30) when the field measurements finished. I am assured that the laboratory experiments of $PM_{2.5}$ and $PM_1$ were conducted in a same batch of experiments including DNA extraction, PCR amplication, real-time qPCR, and Illumina Sequencing except A29 (accidentally omitted in the first batch of Illumina Sequencing). Considering the fact that sequence varied different in different batches of experiments, we have remove the A29 before quality control. A robust statistical analysis of raw sequences were preformed before diversity and taxonomic analysis. After Miseq sequencing, the raw sequences were saved by Fastq files. The Q value (Phred quality score) were calculated by the following equation:

$$Qphred=-10\log_{10}(p)$$

*p indicates the base read error rate

The paired reads were jointed together into sequences by soft FLAST. The quality control were conducted includes: a) removing the primers and barcodes; b) removing the low-quality sequences (length < 250 bp and Q value < 20); c) removing the chimeric sequences. The valid sequences were shown as below.

Table 1 Raw sequences and valid sequences number of samples.

| No. | RS | VS | No. | RS | VS | No. | RS | VS |
|---|---|---|---|---|---|---|---|---|
| A1 | 16770 | 14551 | A21 | 58617 | 51755 | B12 | 43968 | 39322 |
| A2 | 38089 | 32550 | A22 | 45199 | 40554 | B13 | 34925 | 31512 |
| A3 | 100967 | 79898 | A23 | 57862 | 46376 | B14 | 50917 | 44886 |
| A4 | 12236 | 9109 | A24 | 63683 | 50015 | B15 | 72251 | 63627 |
| A5 | 35098 | 20950 | A25 | 16412 | 13938 | B16 | 15817 | 13991 |

| | RS | VS | | RS | VS | | RS | VS |
|-----|--------|--------|-----|--------|--------|-----|--------|--------|
| A6  | 99119  | 51335  | A26 | 43746  | 38228  | B17 | 65677  | 57616  |
| A7  | 82450  | 66653  | A27 | 43877  | 38571  | B18 | 57527  | 52063  |
| A8  | 27325  | 24609  | A28 | 45251  | 38686  | B19 | 77755  | 70479  |
| A9  | 100807 | 47939  | A30 | 180380 | 164935 | B20 | 56931  | 48796  |
| A10 | 48298  | 44184  | B1  | 27627  | 24078  | B21 | 45951  | 38094  |
| A11 | 144435 | 137037 | B2  | 42178  | 38007  | B22 | 60784  | 50330  |
| A12 | 73806  | 65545  | B3  | 76494  | 53373  | B23 | 10202  | 8717   |
| A13 | 123617 | 111296 | B4  | 15338  | 13969  | B24 | 152770 | 127661 |
| A14 | 47854  | 38137  | B5  | 56068  | 50431  | B25 | 48400  | 43593  |
| A15 | 38086  | 32625  | B6  | 13823  | 9797   | B26 | 47504  | 41459  |
| A16 | 100545 | 83655  | B7  | 70444  | 61531  | B27 | 63400  | 56821  |
| A17 | 25850  | 9008   | B8  | 58302  | 51779  | B28 | 50117  | 43514  |
| A18 | 35313  | 31841  | B9  | 17488  | 12427  | B29 | 81316  | 73897  |
| A19 | 61030  | 55692  | B10 | 41966  | 37598  | B30 | 34285  | 29926  |
| A20 | 21763  | 18760  | B11 | 29285  | 26147  |     |        |        |

* RS indicates Raw Sequences number

* VS indicates Valid Sequences number

We have revised as in Page 11  Line 10-15 and Page 12 Line 7-10:

Page 11  Line 10-15:

5 *The remaining filters were analyzed in the same batch of laboratory experiments, including DNA extraction, PCR amplification, quantitative real-time PCR (qPCR), and Illumina sequencing, except for sample A29 in December 9, 2014 (accidentally omitted in the first batch of Illumina sequencing). Considering that a part of the sequences in the 2 batches of experiments differed, we removed this sample before quality control.*

10 Page 12 Line 7-10:

*After high-throughput sequencing, we removed the chimeric and low-quality sequences using the FASTX-ToolKit (http://hannonlab.cshl.edu/fastx_toolkit) and UCHIME algorithm (Edge et al., 2011) before diversity analysis and statistical analysis. The remaining high-quality sequences were normalized to 7973 reads to compare the different samples effectively.*

5. For the guideline values (800 CFU/m3), usually they refer to culturable bacterial CFU, while in their report they detected sequences. For fungal concentration levels, 800 CFU/m3 is a lot higher for most places.

Response of the authors: The guideline (800 CFU m$^{-3}$) was developed for the culturable fungal CFU by Chinese Academy of Sciences Ecological Environmental Research Center. To date, there is no uniform guidelines for the fungal concentration based on the qPCR. So we have deleted the unreasonable comparison with this guideline value (800 CFU m$^{-3}$).

6. Last, some sentences were too verbal, e.g., "got" bigger. What does <typically «l00) mean? Also it is "cultuirng" not cultured" method.

Response of the authors: Thanks for your suggestion, the expression"typically << 100" means"typically less than 100"and " got bigger" means "the particle size increased". I have revised as in Page 8 in Line 26 and Page 9 in Line 24. The remaining verbal sentences were modified by a native English speaker and highlighted in the revised manuscripts.

7. One suggestion to improve their paper is to try differentiate Mt. Tai from ground as a less human impact location (although there are also a lot of visitors). In this way, they might argue that what is fungal level and composition in less polluted higher atmosphere, and further derive potential conclusion about their presence and impact on climate or other things.

Response of the authors: Thanks for your suggestion, we have revised introduction, discussion, and conclusion sections in the revised manuscript.

[revised manuscript text omitted]

---

## Author Response (AR2)

**Response to the Referees**

We thank the reviewers for the beneficial comments on our manuscript. We respond to the reviewer comments in detail below. The responses to reviewer are in red.

**Comments from Anonymous Referee #3**

5   1, The authors need to clearly give the meaning of $PM_{2.5}$ and $PM_1$ they used in this study in METHOD (for example, by showing the details of the samplers) and in ABSTRACT. As described in the manuscript, the samplers they used to collect the samples were inertial impactors. With such impactors, particles smaller than a certain aerodynamic size could not be trapped by the collection filters. So the samples of what the authors called $PM_{2.5}$ and $PM_1$

10   corresponded to size ranges of 2.5 - ? um, and 1.0-? um, respectively. That means the $PM_{2.5}$ and $PM_1$ in this study are different from the general definition of $PM_{2.5}$ and $PM_1$. I guess this is also the reason for the results that the concentration of fungi in $PM_1$ was sometimes larger than the concentration in $PM_{2.5}$. This is because a particle of geometric diameter larger than 1.0 um might have an aerodynamic diameter smaller than 1.0 um. The fungi in this study were likely the case. Particle bounce is usually small on quartz filters, unless the particles are

15   much larger than the gaps or holes between quartz fibers on the filter surface.

Response of the authors:

Thanks for your suggestion. Yes, the samplers were deployed with particles larger than 2.5 μm and 1μm trapped by the impactors and particles smaller than 2.5 μm and 1 μm collected

20   on the quartz filters, respectively. The 50% cutoff aerodynamic diameter are 2.5 μm and 1 μm, respectively. We have added the details of the sampling instruments in Method and Abstract as in Page 7 Line 4-8 and Page 4 Line 22-23.

Page 4 Line 22-23: fine ($PM_{2.5}$, 50% cutoff aerodynamic diameter $D_{a50}$=2.5 μm, geometric standard deviation of collection efficiency $\sigma_g$=1.2) and submicron ($PM_1$, $D_{a50}$=1 μm, $\sigma_g$=1.2)

25   Page 7 Line 4-8: Two middle-volume (100 L min$^{-1}$) samplers (TH-150A; Wuhan Tianhong Instruments Co. Ltd., Wuhan, China) were deployed with particles larger than 2.5 μm and 1μm trapped by the impactors and particles smaller than 2.5 μm and 1 μm collected on the quartz filters, respectively. The 50% cutoff aerodynamic diameter are 2.5μm and 1μm, respectively. The smaller the aerosol particles, the higher collection efficiency.

30   Page 10 Line 6-7: There is no significant differences between $PM_{2.5}$ and $PM_1$ based on the uncertainty estimate (95% confidence intervals).

2, The authors need to give the DNA extraction efficiency in subsection 2.2. The major contribution of this study is the quantification of fungi in the air. Quantifying the fungi is different from the identification of the fungi presence. If the efficiency is unknown, the concentrations reported in the paper are not referentially meaningful and hardly compared with other studies. I know it is hard to give an exact efficiency. However, the efficiency or at least the merits and limits of the extraction method used in this study, in comparison with methods in other studies, should be described.

Response of the authors: Thanks for your suggestion. We have added the details about the DNA extraction efficiency in subsection 2.2 in Page 7 Line 29-34:

The sample pretreatment and DNA extraction experiments were performed following an protocol optimized by Jiang et al., (2015). This protocol can extract sufficient DNA from low-biomass environmental samples (e.g., aerosol particles, and other alike) and boosted the DNA extraction efficiency more than twice better than the non-optimized extraction method. Besides, it has been applied for studying airborne microbial diversity in different environment (Cao et al., 2014; Deng et al., 2016; Tong et al., 2017; and Gao et al., 2017b).

**Comments from Anonymous Referee #4**

Primary biological aerosol particles (PBAPs), including pollen, bacteria, fungi, are the important component of ambient particles in atmosphere. PBAPs not only play unique role in

global climate change, but also take negative effects on human health. As one important species, fungi (with big size, coarse particles) could eject their spore (with smaller sizes, fine or ultrafine particles) with aqueous jets into atmosphere and therefore bring more public health concerns. This manuscript focuses on fungal distribution in $PM_1$ and $PM_{2.5}$ collected in Mountain Tai. Cutting-edge methods (such as sequence analyses) were employed to deterimine species of fungal, and especially, the authors calcualte fungal contribution to the ambient OC.

Generally speaking, the authors could give the aim of their study and design their experiment carefully and describe their results clearly. The reviewer believes this manuscript is of the interest for the readers of ACP and therefore suggests it can be considered for publication before the authors pay attention on the following minor problems:

Response of the authors: We thank the reviewer for the beneficial comments on our manuscript.

1, Plesase consider the implication of the allergenic or pathogenic fungal in the discussion as a indepent section. The author mentioned the health problems in the section 3.3

Response of the authors: Thanks for your suggestion. We have separated this paragraph into section 3.4.

2. Could the authors please give explanation why you have not gotten surface ground samples, becuase you compared the data in mountain Tai with that in Korea, Austria. The reviewer think fungal distribution at different height of the Mt. Tai must be more interesting.

Response of the authors: This study focused on the ambient fungi at an elevated site in the North China Plain. Diverse microbes at high altitudes (such as in cloud water and precipitation) can act as nucleating agents for cloud and ice condensation, influence precipitation patterns, and drive the biogeochemical cycling of elements in ecosystem processes. But the similar researches performed at high mountains were few, wherefore we can only compare the results with that in surface ground samples such as Korea, Austria.

3. in the page 10 line 35, "differ" should be "difference". And the conclusion could be more concise if the first sentence " Information about ........" was deleted.

Response of the authors: We have revise 'differ' to 'difference' in Page 14 Line 20 and deleted the 'Information about' in Page 14 Line 14.

[revised manuscript text omitted]